# Ex vivo editing of human hematopoietic stem cells for erythroid expression of therapeutic proteins

Giulia Pavani[1], Marine Laurent [1], Anna Fabiano [1], Erika Cantelli[1], Aboud Sakkal[1], Guillaume Corre [1], Peter J. Lenting [2], Jean-Paul Concordet[3], Magali Toueille[1], Annarita Miccio[4,5] & Mario Amendola [1✉]

Targeted genome editing has a great therapeutic potential to treat disorders that require protein replacement therapy. To develop a platform independent of specific patient mutations, therapeutic transgenes can be inserted in a safe and highly transcribed locus to maximize protein expression. Here, we describe an ex vivo editing approach to achieve efficient gene targeting in human hematopoietic stem/progenitor cells (HSPCs) and robust expression of clinically relevant proteins by the erythroid lineage. Using CRISPR-Cas9, we integrate different transgenes under the transcriptional control of the endogenous α-globin promoter, recapitulating its high and erythroid-specific expression. Erythroblasts derived from targeted HSPCs secrete different therapeutic proteins, which retain enzymatic activity and cross-correct patients' cells. Moreover, modified HSPCs maintain long-term repopulation and multilineage differentiation potential in transplanted mice. Overall, we establish a safe and versatile CRISPR-Cas9-based HSPC platform for different therapeutic applications, including hemophilia and inherited metabolic disorders.

[1] INTEGRARE, Genethon, UMR_S951 Inserm, Univ Evry, Univ Paris-Saclay, 91002 Evry, France. [2] Laboratory of Hemostasis-Inflammation-Thrombosis, UMR_S1176 Inserm, Univ. Paris-Sud, Université -Saclay, 94276, Le Kremlin-Bicêtre, Orsay, France. [3] National Museum of Natural History, UMR_1154 Inserm, UMR_7196 CNRS, Univ Sorbonne, Paris, France. [4] Imagine Institute, UMR_163 INSERM, Paris, France. [5] Paris Descartes Univ Sorbonne Paris Cité, Paris, France. ✉email: mamendola@genethon.fr

Many diseases require protein replacement therapy (PRT) to supplement a protein that is deficient because of a genetic defect. PRT is approved or under investigation to treat more than 40 inherited disorders, mostly involving blood factors and lysosomal enzymes[1]. Although life saving for some patients, this therapy has several limitations, that lead to treatment failures and limited long-term efficacy[2].

Genome editing technologies have a great therapeutic potential for genetic disorders, as they can fix the underlying disease-causing mutation[3,4]. However, this approach requires the development of countless gene-tailored editing strategies that can hinder clinical translation.

To overcome this issue, a single "safe harbor" or a highly transcribed genomic locus can be exploited to integrate and overexpress different therapeutic transgenes[5]. Previous studies successfully used adeno-associated virus (AAV) for nuclease-mediated targeting of transgenes under the control of the endogenous albumin promoter in liver[6,7]. The striking transcriptional activity of this locus achieved therapeutic protein levels in different preclinical models and thus prompted the first in vivo genome editing trial in humans (NCT03041324). Although promising, this approach is hampered by: (1) presence of pre-existing antibodies against AAV capsid that precludes treatment to a significant portion of patients[8]; (2) long-term expression of synthetic nucleases in vivo, which could result in genotoxicity and trigger immune responses against transduced hepatocytes[9,10]; (3) liver conditions that can alter AAV transduction and hepatic protein expression[11,12].

As therapeutic alternative, hematopoietic stem cells (HSCs) can be harnessed to overexpress transgenes in downstream hematopoietic lineages. Differently from liver, autologous HSCs can be easily accessed for ex vivo gene manipulation and re-administration, thus circumventing immunological issues; however, a suitable locus for transgene integration (knock-in, KI) still needs to be identified.

α-globin genes are expressed by the erythroid lineage at extremely high levels (~1.5 g/day)[13], they are present in 4 copies per cells and the loss of up to 3 α-globin alleles is mostly asymptomatic[14], making this locus a promising candidate for KI in HSCs. In addition, erythroid cells are the most abundant hematopoietic progeny (~$2 \times 10^{11}$ new erythrocytes per day)[13] and can secrete relevant amounts of therapeutic proteins, as previously demonstrated by gene transfer using lentiviral vectors (LV)[15–17].

Here, using CRISPR-Cas9 we integrate therapeutic genes under the transcriptional control of the endogenous α-globin promoter in human HSCs. We aim to combine strong transcription and abundance of transgene-expressing erythroblasts to maximize protein production, reducing the number of integration events required to reach therapeutic levels.

This HSC platform for robust erythroid-specific expression of therapeutic proteins opens possibilities for treating hemophilia and lysosomal storage disorders (LSD), as well as other genetic diseases.

## Results

**Selection of gRNA targeting the α-globin locus**. To generate a DNA double-strand break (DSB) for transgene integration in the α-globin locus, we focused on *Streptococcus pyogenes* (*Sp*)Cas9 nuclease, the only Cas in clinical trials to edit HSCs (NCT03164135; NCT03655678). We designed 14 guide (g)RNAs targeting the non-coding sequences of α-globin genes, in particular the 5′ untranslated region and introns (5′UTR, IVS1 and IVS2 respectively), avoiding known regulatory elements (Fig. 1a and Supplementary Table 1A). gRNA were tested for on-target

DNA cleavage (InDels) in K562 erythroleukemic cells constitutively expressing Cas9 (Fig. 1b). For the best candidates for each region we analyzed the indels pattern (Supplementary Fig. 1a) and we assayed their effect on α-globin production. As control, we designed a gRNA (KO) targeting the first exon of *HBA1* and *HBA2* genes, which abrogates α-globin production. In K562, 5′UTR and IVS2 gRNA did not alter α-globin protein level (Supplementary Fig. 1b) and were therefore selected for further investigation.

To evaluate DNA cleavage efficiency in clinically-relevant human HSPCs, cells were transfected with Cas9/gRNA ribonucleoprotein complex (RNP). To control the effects of the editing procedure, we included a gRNA targeting an unrelated genomic locus (AAVS1). We observed efficient editing for 5′UTR and IVS2 gRNA in both erythroid liquid culture and methylcellulose-plated colony-forming cells (CFC) (Fig. 1c), which did not affect HSPC viability and multilineage potential (CFC assay; Fig. 1d and Supplementary Fig. 1c) or altered erythroid differentiation (flow cytometry analysis, Fig. 1e). Remarkably, 5′UTR and IVS2 gRNA did not modify α-globin expression, measured as ratio between α- and β-like globin chains (Fig. 1f and Supplementary Fig. 1d). In accordance with these data, adult hemoglobin (2α+2β globin chains; HbA) remained the predominant hemoglobin form in both 5′UTR and IVS2 erythroblasts, while it strongly decreased in KO controls where alternative homotetramers lacking α-globin chains appeared, as in α-thalassemic patients[14] (Fig. 1g and Supplementary Fig. 1e). Lastly, since the two α-globin genes (*HBA1* and *HBA2*) are the result of evolutionary duplication (96.67% sequence homology, GRCh38), we evaluated if simultaneous cleavage of both genes can induce loss of *HBA2* in edited HSPCs. We observed a reduction of *HBA2* copies per cell to 1.8 ± 0.3 for IVS2 gRNA, which selectively targets *HBA2*, and to 1.4 ± 0.3 for 5′UTR gRNA (Supplementary Fig. 1f); however, these rearrangements had minimal effect on globin production, as shown above. Detection and quantification of *HBA2* inversions was not possible due to technical issues associated with the presence of repetitive sequences and the high GC content of the α-globin locus.

Overall, these results demonstrate that both 5′UTR and IVS2 gRNA efficiently cut α-globin genes without affecting HSPC viability, differentiation potential and hemoglobin expression, thus representing an interesting genomic locus to test KI.

**Targeted integration**. To evaluate if the α-globin promoter can drive the expression of an integrated heterologous transgene, we generated KI cassettes containing a promoterless GFP (Supplementary Fig. 2a). These cassettes were delivered in K562-Cas9 cells using integrase-defective lentiviral vector (IDLV) and integrated by transfecting a gRNA encoding plasmid. Interestingly, all gRNA/IDLV combinations resulted in GFP expression, which increased upon erythroid differentiation (Supplementary Fig. 2b). In addition, on-target integration by non-homologous end joining was confirmed in GFP positive clones by PCR (Supplementary Fig. 2c) and the presence of a chimeric messenger RNA showed correct splicing of intron traps (Supplementary Fig. 2d). Similar results were obtained upon KI in the β-globin gene, suggesting that KI in globin genes with different expression levels could be a viable strategy to modulate transgene expression (Supplementary Fig. 2e–j).

We further confirmed these α-globin KI data in immortalized human erythroid progenitor cells (HUDEP-2)[18], which can differentiate to reticulocytes. To perform KI, HUDEP-2 cells were transfected with 5′UTR or IVS2 RNP and transduced with an AAV6 carrying the aforementioned expression cassettes flanked by homology arms to favor homologous DNA recombination

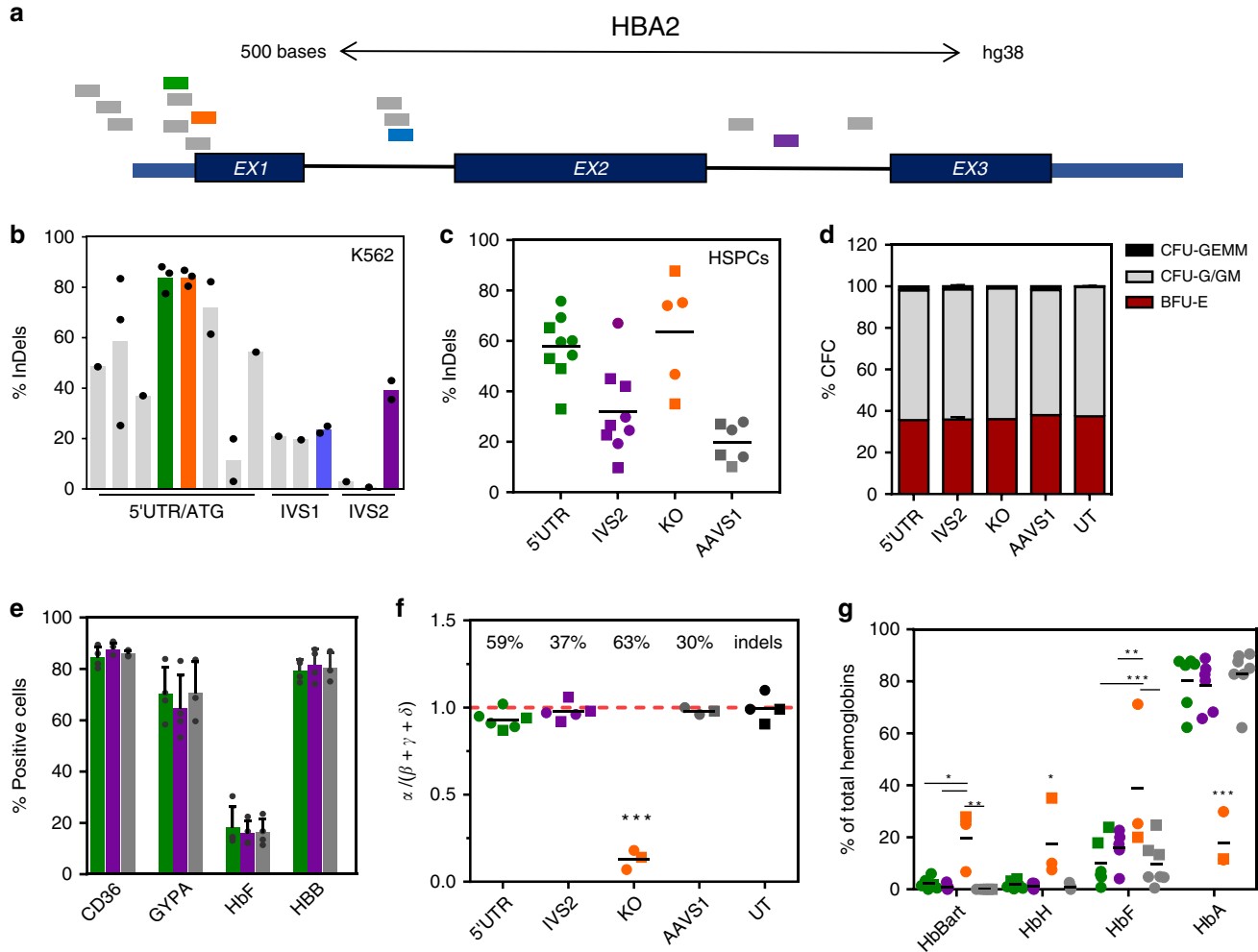

**Fig. 1 Editing of selected sites in the α-globin locus minimally affects globin production. a** Locations of gRNA on HBA2 gene. All gRNA except 74 (in purple) target both HBA1 and HBA2. Selected gRNA are highlighted. **b** K562-Cas9 screening of gRNA targeting different region of the α-globin locus (5′ untranslated region/start codon (5′UTR/ATG), intron 1 (IVS1) or intron 2 (IVS2)). Each bar is a different gRNA, each dot a different experiment. Editing efficiency is expressed as percentage of modified HBA alleles (bars represent mean; data from 1, 2 or 3 biological replicates). The gRNA selected for each region are highlighted. **c** Editing efficiencies in HSPCs in erythroid liquid culture (circles) or in BFU-E (burst-forming unit-erythroid, squares). Lines show mean (2-4 donors; $n = 5$ KO, $n = 6$ AAVS1, $n = 9$ 5′UTR and IVS2). **d** Colony-forming cell (CFC) frequency in edited HSPCs (mean ± SD; $n = 2$, $n = 4$ for IVS2). CFU-GEMM, granulocyte, erythroid, macrophage, megakaryocyte; BFU-E, burst-forming unit-erythroid; CFU-G/GM, granulocyte-macrophage. **e** Flow-cytometry analysis of erythroid markers upon differentiation of edited HSPCs, day12 (green 5′UTR, purple IVS2 and gray AAVS1). Results are shown as mean ± SD ($n = 3$ biological replicates, 3 different donors; $p = 0.94$, one-way ANOVA Tukey's test). **f** HPLC analysis of hemoglobin monomers of erythroblasts derived from edited HSPCs ($n = 6$ 5′UTR, $n = 5$ IVS2, $n = 3$ KO and AAVS1, $n = 4$ UT; 4 donors). InDels percentage mean is indicated. The ratio α/β-like globins in normal cells is close to 1 (red dashed line). Black lines show mean; BFU-E (squares), erythroid liquid culture (circles) (***$p < 0.001$ vs UT, 5′UTR, IVS2, AAVS1; one-way ANOVA, Tukey's test). **g** HPLC analysis of hemoglobin tetramers of erythroblasts derived from edited HSPCs (same samples as in **f**). Every tetramer is reported as % of total hemoglobins (*$p < 0.05$; **$p < 0.01$; ***$p < 0.001$; two-way ANOVA, Dunnet's test). Black lines show mean; BFU-E (squares), erythroid liquid culture (circles). Hb Bart, $\gamma_4$ ($p = 0.017$ 5′UTR, $p = 0.013$ IVS2, $p = 0.006$ AAVS1 vs KO); HbH, $\beta_4$ ($p = 0.038$ 5′UTR, $p = 0.035$ IVS2, $p = 0.032$ AAVS1 vs KO); HbF, fetal hemoglobin, $\alpha_2\gamma_2$ ($p < 0.001$ 5′UTR and AAVS1, $p = 0.002$ IVS2 vs KO); HbA, adult hemoglobin, $\alpha_2\beta_2$ ($p < 0.001$ vs KO). Source data are in the Source Data file.

(HDR)[19] (Fig. 2a). After puromycin selection, GFP was expressed from both genomic sites and increased about 100 fold upon differentiation, with 5′UTR integration expressing ~10 fold higher than IVS2 (Fig. 2b, c).

We then performed 5′UTR and IVS2 KI in HSPCs, where HDR was more efficient and no enrichment was required. Again, we observed a similar GFP upregulation after erythroid induction (~100 fold) and a higher expression upon 5′UTR integration (Fig. 2d–f and Supplementary Fig. 2k).

For this reason and considering that DNA targeted integration in IVS2 could result in the expression of a truncated α-globin chain, we selected the 5′UTR region for further investigation.

Importantly, even if PCR analysis of individual CFC showed integration in both erythroid and granulocyte-monocyte colonies (Supplementary Fig. 2l), flow cytometry and microscopy data demonstrated that GFP expression was restricted to erythroid progenitors (Fig. 2g, h). Sanger sequencing of PCR products spanning the AAV-genome junction of colonies showed that KI occurred through HDR ($n = 10$; Supplementary Fig. 2l, m) Taken together, these data show that KI into α-globin locus is efficient and results in robust erythroid-specific expression.

**Off-targets analysis**. We performed off-target analysis for 5′UTR gRNA in K562. Briefly, cells were transfected with RNP for 5′

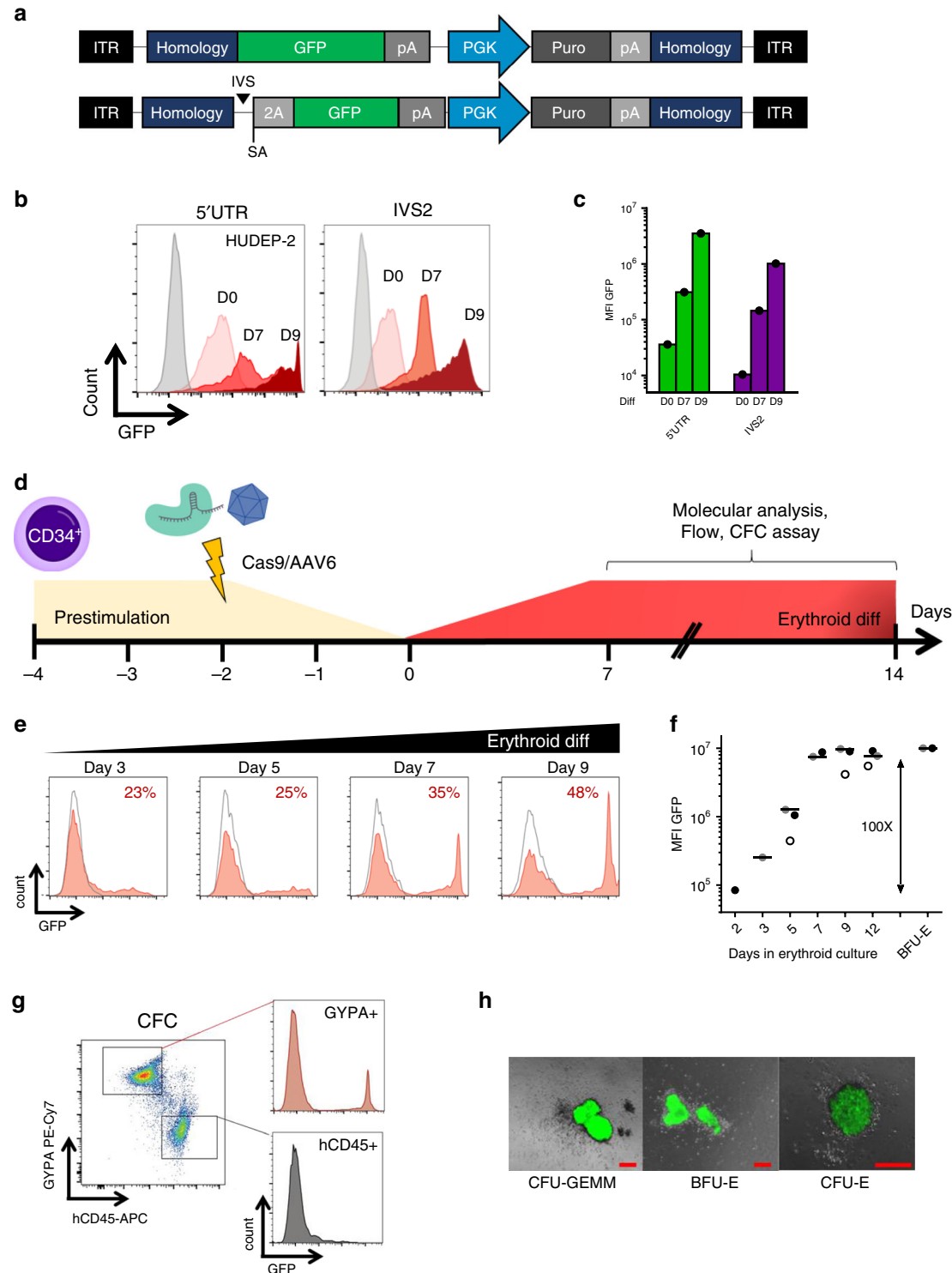

UTR or AAVS1 (negative control), and 29 top-scoring off-targets predicted in silico[20,21] were PCR amplified for InDels quantification. Although on-target activity reached >85%, we could not detect any difference between HBA15 and control AAVS1 gRNA at any of the predicted HBA15 off-target sites (Supplementary Table 2) (with a technical threshold of >2% of TIDE software)[22]. To detect unpredicted off-targets, we also performed an unbiased genome-wide screening by IDLV capture analysis. This method exploits the tendency of IDLV genome to ligate into DSB, thereby stably tagging otherwise undetectable DSB[23,24]. To this purpose, K562 were transduced in triplicate with an IDLV encoding for a

constitutively expressed GFP and transfected with 5′UTR or IVS2 RNP. The latter sample was used as positive control, since IVS2 gRNA has a known off-target in *HBA1*. After GFP-based sorting, we identified IDLV clustered integration sites (CLIS) by ligation mediated (LM)-PCR coupled with next generation sequencing[25]. As expected, we observed high on-target activity for both gRNA (92.5% and 91.2% of total reads for 5′UTR and IVS2 gRNA, respectively), and we confirmed integration in the predicted *HBA1* off-target for IVS2 (8.5%), in line with TIDE-based indels analysis (8.5% ± 4.9, *n* = 5) (Supplementary Fig. 3a–c). Remarkably, no unique CLIS and none of the predicted off-targets were

**Fig. 2 Transgene integration into the α-globin locus results in robust erythroid-specific expression. a** AAV6 donors used for KI experiments in 5′UTR (top) and IVS2 (bottom) of the α-globin genes. Both vectors contain a promoterless GFP with bovine growth hormone polyA (pA), followed by a phosphoglycerate Kinase (PGK) promoter with a puromycin selection marker (puro) and simian virus polyA (pA). This cassette is flanked by 250 bp homology arms (homology) to gRNA target. IVS2 trap also contains a synthetic intron (IVS), a splice acceptor (SA) and a self-cleaving peptide (2A). ITR, Inverted terminal repeats. **b** Representative histograms of GFP expression of HUDEP-2 KI cells at day 0 (light pink), day 7 (red) and day 9 of erythroid differentiation (dark red). Untreated HUDEP-2 are shown in gray ($n = 1$). **c** Barplot of GFP median fluorescent intensity (MFI) as in **b**. **d** Schematic representation of HSPC targeting experiments. **e** Representative histograms of GFP expression of 5′UTR KI (red fill) and AAV6 only HSPCs (gray line) during erythroid differentiation. Percentage of GFP positive cells is indicated ($n = 3$ different donors). **f** GFP median fluorescent intensity (MFI) during differentiation of 5′UTR KI HSPCs (lines indicate mean, $n = 3$ different donors indicated by open, gray and black circles). **g** Representative dot plots showing GFP expression in erythroid (GYPA+) and leukocytes (hCD45+) CFC. **h** Representative overlay images (bright field and GFP channel) of different erythroid progenitor-derived colonies ($n = 24$). Scale bars in red indicate 200 μm. CFU-GEMM, granulocyte, erythroid, macrophage, megakaryocyte; BFU-E, burst-forming unit-erythroid; CFU-E, erythroid. Source data are in the Source Data file.

identified for 5′UTR gRNA after correction for random IDLV integration, further assuring the lack of any predominant off-target for this gRNA.

**Hemophilia B**. As first therapeutic target, we tested our platform for hemophilia B (OMIM #306900), a disease model for gene-based therapies caused by the absence of functional Factor IX (FIX, F9). Initially, HUDEP-2 cells were transfected with 5′UTR RNP and transduced with an AAV6 carrying two 250 bp homology arms flanking a promoterless human FIX-R338L (FIX Padua[26]) and a constitutive GFP reporter to easily track KI cells (Fig. 3a). Concordance between DNA integration and GFP expression analyses before and after GFP sorting confirmed that most integrations were on-target (Fig. 3b), with a preference for *HBA1* integration (Supplementary Fig. 4a). FIX expression was upregulated upon HUDEP-2 erythroid differentiation (Fig. 3c) and its secretion (median 1161 ng/10⁶ cells/FIX copy, 769.1-1885, interquartile range) correlated with the number of integrated FIX copies (Fig. 3d).

Editing of HSPCs showed that also in primary cells, without any selection, we could obtain high levels of InDels (Supplementary Fig. 4b) and KI of FIX as measured by GFP (Fig. 3e) and on-target ddPCR (Supplementary Fig. 4c), associated with a reduced number of *HBA2* copies (Supplementary Fig. 4d).

Once more, we could demonstrate that *F9* mRNA and protein secretion increased upon erythroid differentiation (Fig. 3f, g) and that secreted FIX was functional (Fig. 3h; Supplementary Fig. 4e). Interestingly, FIX expression achieved with targeted integration was higher compared to a state-of-the-art LV carrying an artificial β-globin promoter[27] (Fig. 3l, j), highlighting one of the advantages of exploiting endogenous promoters in their chromatin context. Analysis of HSPC derived colonies, confirmed that high KI efficiency in CFC (both erythroid and granulocyte-monocyte colonies, Supplementary Fig. 4f) did not affect HPSC clonogenic differentiation capacity (Supplementary Fig. 4g), although the total number of CFC was lower than control HSPCs due to known AAV toxicity (Supplementary Fig. 4h)[28,29]. In addition, by analyzing KI HSPC derived burst-forming unit-erythroid colonies (BFU-E) we showed that *F9* integrations were mostly monoallelic (Fig. 3k) and HDR-mediated (19/19 colonies; Supplementary Fig. 4i), associated with a reduced number of *HBA2* copies (Supplementary Fig. 4l). Importantly, also BFU-E derived erythroblasts were capable of secreting FIX (Supplementary Fig. 4m).

These results clearly indicate that this platform can express and secrete a functional protein with therapeutic relevance.

**Lysosomal storage disorders**. In light of these promising findings, we expanded our strategy to other genetic diseases eligible for PRT, such as LSD. These inherited metabolic conditions are characterized by an abnormal build-up of toxic metabolites in

lysosomes as a result of enzyme deficiencies[30]. Here we tested three different human transgenes encoding for: α-L-iduronidase (IDUA; Hurler syndrome, OMIM #607014), α-galactosidase (GLA; Fabry disease, OMIM #301500) and lysosomal acid lipase (LAL; Wolman disease, OMIM #278000). To facilitate their detection, each enzyme was tagged with 3 copies of hemagglutinin epitope (HA) and cloned into AAV6 vectors (Fig. 4a). As for *F9*, these transgenes were integrated into the α-globin locus of HUDEP-2 and KI cells were enriched by GFP sorting. Both mRNA and protein analyses confirmed enzymes expression, which substantially increased upon erythroid differentiation (16–171 fold and 2.5–4.5 fold respectively, Fig. 4b, c). For additional experiments in HSPCs we focused on *LAL* transgene, since Wolman disease (WD) is a life-threatening genetic condition with a severe liver phenotype and no gene therapy options available.

Editing of HSPCs showed that, without any selection, we could obtain high levels of InDels (Supplementary Fig. 5a) and KI of *LAL* as measured by GFP (Fig. 4d) and on-target ddPCR (Supplementary Fig. 5b), associated with a reduced number of *HBA2* copies (Supplementary Fig. 5c). In addition, LAL enzyme was strongly expressed and secreted upon erythroid differentiation (Fig. 4e, f) and retained its hydrolytic activity, in accordance with antigen levels (Fig. 4g).

By analyzing KI HSPC derived burst-forming unit-erythroid colonies (BFU-E) we showed that *LAL* integrations were mostly monoallelic (Fig. 4h), associated with a reduced number of *HBA2* copies (Supplementary Fig. 5d). After aggregation of the genotypes of *F9* and *LAL* BFU-E, we established that most of edited BFU-E (87%) had transgene integration and/or *HBA2* deletion and 53% harbored both modifications (Supplementary Fig. 5e, f).

In order to be therapeutically relevant, secreted LAL enzyme should cross-correct LAL deficient cells and reduce pathological cholesterol accumulation in lysosomes. Thus, we exposed WD patient's fibroblasts to conditioned medium from untreated (UT) or KI HSPC derived erythroblasts (LAL). After 3 days we observed LAL uptake in WD fibroblast lysates (Fig. 4i), which correlated with a significant decrease of total cholesterol (Fig. 4j) and lipid deposits (Fig. 4k), clearly showing that the secreted enzyme can ameliorate the metabolic dysfunction. Altogether, we demonstrated that our platform is versatile and can express several functional therapeutic proteins that require post-translational modifications.

**In vivo long-term analysis of edited HSPCs**. To evaluate if LAL-KI HSPCs maintain their homing, engraftment and multi-lineage potential, we transplanted immunodeficient NOD/SCID/γ[31] (NSG) mice and monitored human cells for 16 weeks (Fig. 5a). All mice showed successful engraftment in bone marrow, spleen and blood (Fig. 5b). GFP positive cells were present at different time points (Fig. 5c, d; Supplementary

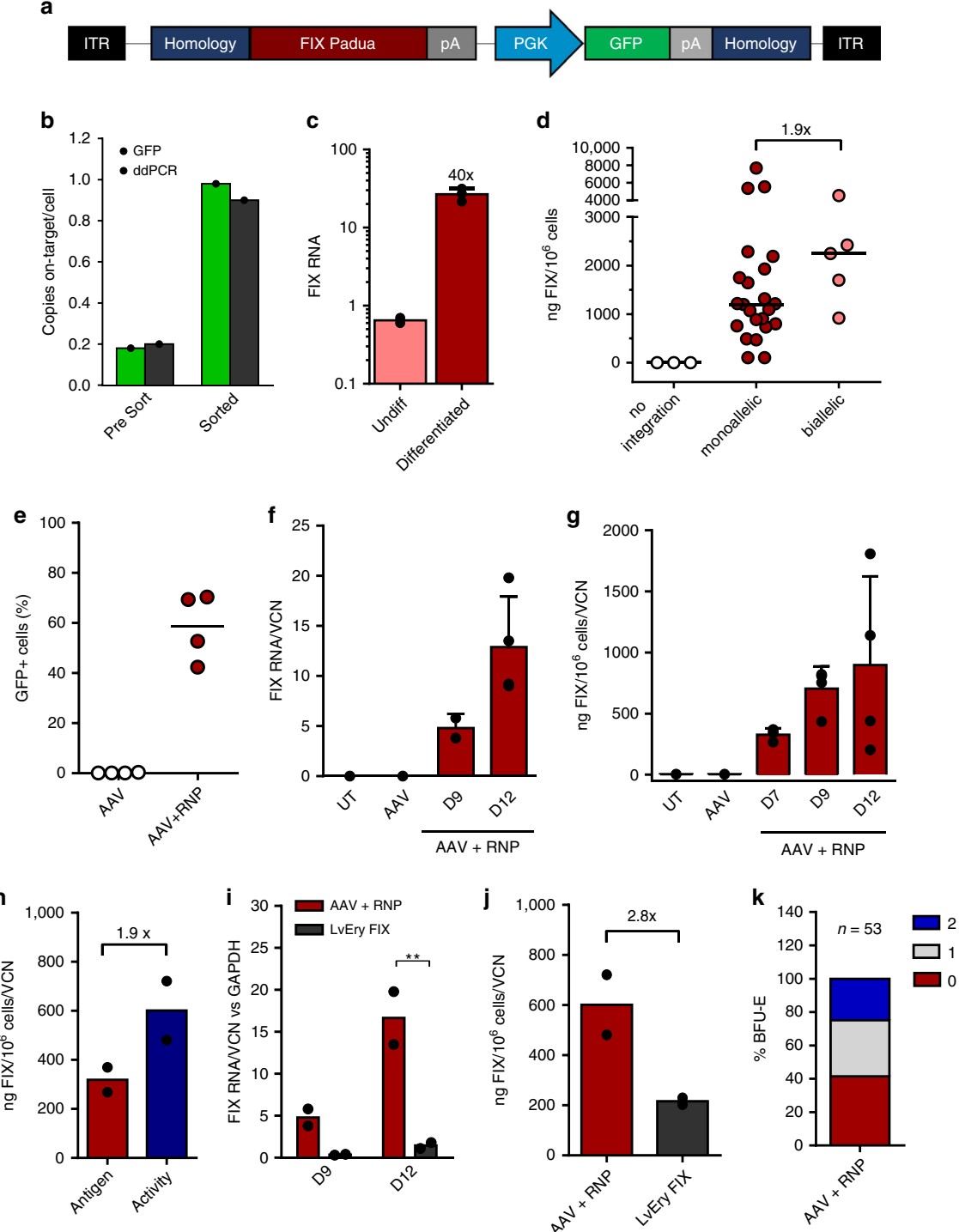

**Fig. 3 *F9* KI into the α-globin locus results in expression and secretion of functional enzyme. a** AAV6 donor used for KI experiments of FIX Padua. **b** FIX KI efficiency in HUDEP-2 cells was measured by flow cytometry (light green) or ddPCR specific for on-target integration (dark green) before and after sorting ($n = 1$). **c** Quantification of FIX mRNA in KI HUDEP-2 upon differentiation (mean ± SD, $n = 2$ undifferentiated, $n = 3$ differentiated). **d** Quantification of FIX secretion in medium of HUDEP-2 clones ($n = 28$) with monoallelic or biallelic KI (ELISA), as detected by on target ddPCR analysis (AAV-genome junction amplification). Lines represent median. **e** KI efficiency in HSPCs at day 9 of erythroid differentiation. Lines represent mean ($n = 4$). **f, g** FIX expression during HSPC differentiation at RNA (**f**, qPCR; $n = 2$ day 9; $n = 4$ day 12) and protein level (**g**, ELISA on supernatants, $n = 3$ day 7; $n = 4$ day 9 and 12; 3 donors). Bars represent mean ± SD. **h** Comparison of FIX antigen (ELISA) and activity (aPTT) in supernatants of KI HSPCs (mean; $n = 2$). **i, j** Comparison of FIX RNA at day 9 and 12 of erythroid differentiation (**i**) and protein (**j**) in KI HSPCs (AAV + RNP) vs HSPCs transduced with an erythroid-specific lentiviral vector (LvEry FIX). Bars represent mean (**p = 0.003 *t*-test Holm-Sidak correction for RNA at day 12; $p = 0.08$ for protein, $n = 2$). **k** Integration pattern in single BFU-E (2 donors): no integration (0), monoallelic (1) and biallelic KI (2). Source data are in the Source Data file.

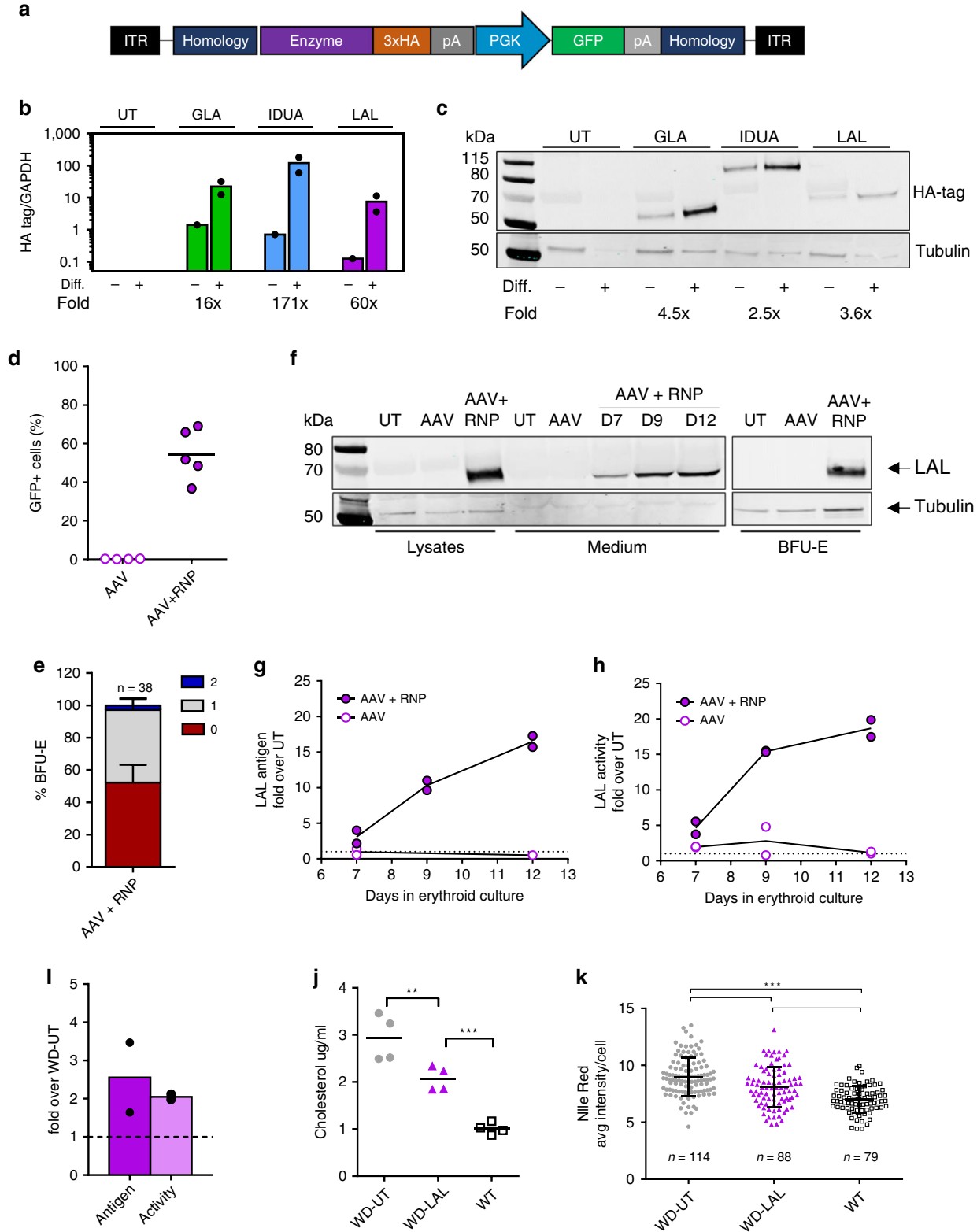

Fig. 6a) and in all cell subsets analyzed (Fig. 5e), including more primitive HSPCs in the bone marrow (Fig. 5f), demonstrating that KI HSCs were able to reconstitute the entire hematopoietic system. However, in accordance with previous reports describing AAV toxicity in HSPCs[28,29], KI HSPCs showed lower engraftment levels compared to unedited HSPCs (Fig. 5b) and a reduction of KI GFP positive cells after transplantation (Fig. 5c).

Since NSG mice do not support human erythroid differentiation[32], we isolated human CD34[+] cells from bone marrow of engrafted mice and differentiated them ex vivo. In a CFC assay, KI HSPCs were still able to generate both erythroid and myeloid colonies, to express GFP (Supplementary Fig. 6b) and, most importantly, to produce LAL in erythroblasts (Fig. 5g and Supplementary Fig. 6c). Similar in vivo and ex vivo results were also obtained for FIX (Supplementary Fig. 6d–g).

**Fig. 4 Expression and therapeutic potential of different lysosomal enzymes. a** AAV6 donor used for KI experiments. All enzymes were tagged with hemagglutinin tag (3xHA). **b** Transcript upregulation of different enzymes in targeted HUDEP-2 upon differentiation (qPCR, $n = 2$, mean). Fold increase is indicated. **c** Representative western blot detecting different enzymes (HA-tag and anti-β tubulin) of targeted HUDEP-2 upon differentiation ($n = 2$). **d** KI efficiency of LAL-AAV6 in HSPCs at day 9 (lines indicate mean; AAV $n = 4$, AAV + RNP $n = 5$). **e** Representative western blot of LAL in HSPC lysates, supernatants and BFU-E in untreated (UT), transduced (AAV) and KI-HSPCs (AAV + RNP). Anti-HA tag and anti-β tubulin antibodies were used.
**f** Quantification of secreted LAL during erythroid differentiation. Anti-LAL antibody was used. Data are shown as fold increase over untreated cells (UT, donor=2). **g** LAL activity in HSPC supernatants during erythroid differentiation, data are shown as fold increase over untreated cells (UT, $n = 2$).
**h** Integration pattern in single BFU-E: no integration (0), monoallelic (1) and biallelic (2). Mean ± SD, donor = 2). **i** Uptake of erythroid-expressed LAL by WD fibroblasts, measured by western blot or activity assay (mean; $n = 2$). **j** Cholesterol levels in WD fibroblasts after incubation with conditioned medium from untreated (UT) or LAL KI-erythroblasts. WT fibroblasts are shown as control ($n = 4$; $p = 0.003$ WD-UT vs WD-LAL; $p < 0.001$ WD-LAL vs WT; one-way ANOVA, Tukey's test). **k** Nile Red staining in WD fibroblasts after incubation with conditioned medium from untreated (UT) or LAL KI-erythroblasts. WT fibroblasts are shown as control. Black lines indicate mean ± SD; number of fibroblasts analyzed is indicated. (***$p < 0.001$ one-way ANOVA, Tukey's test). Source data are in the Source Data file.

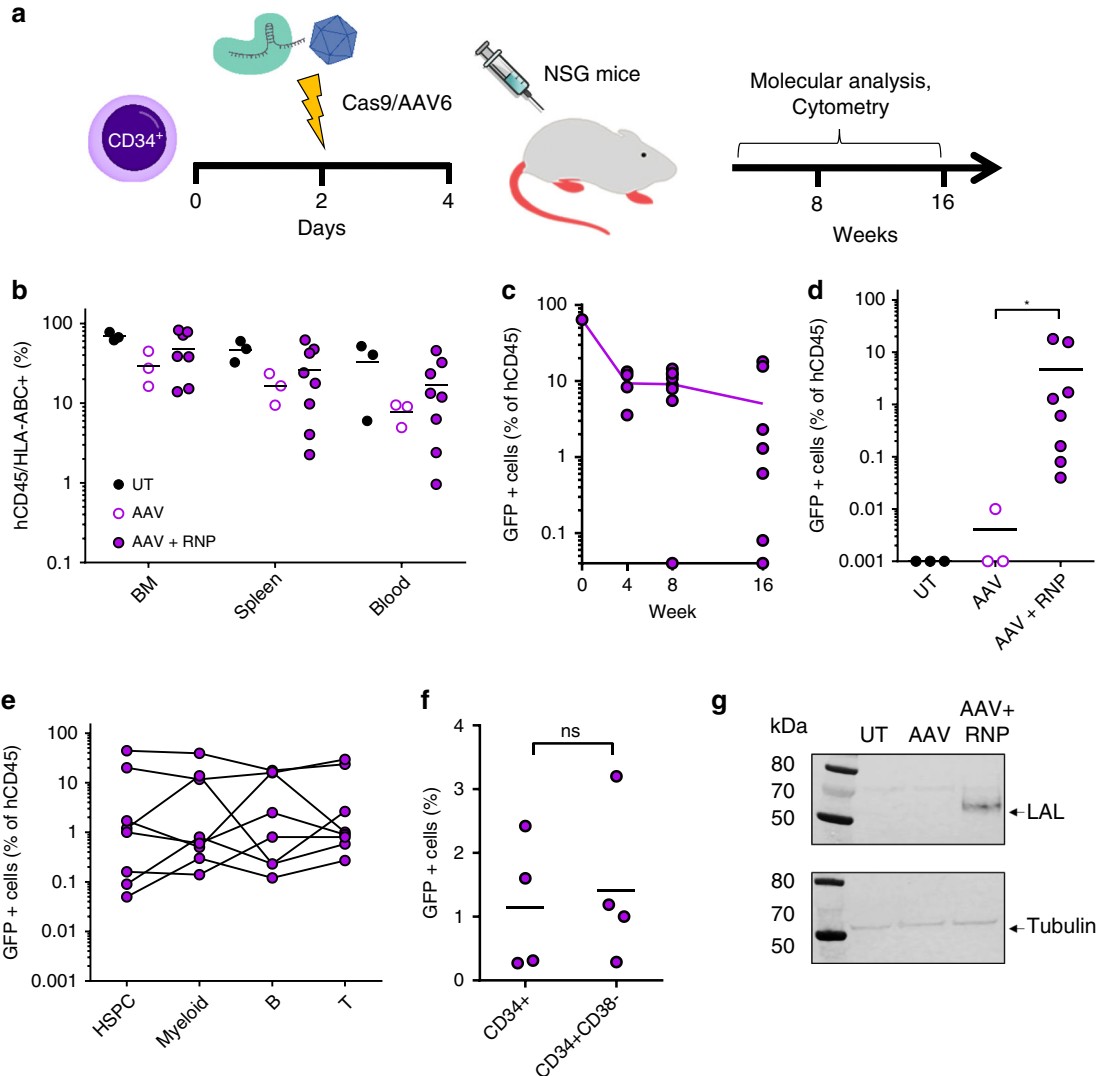

**Fig. 5 KI HSPCs engraft NSG mice and express LAL upon erythroid differentiation. a** Schematic representation of engraftment experiments. **b** Percentage of human CD45+/HLA-ABC+ cells in hematopoietic organs of mice. BM = bone marrow. (UT and AAV $n = 3$; AAV + RNP $n = 8$). **c** GFP positive cells in peripheral blood of transplanted mice over time. Line indicates mean ($n = 4$ week 4; $n = 8$ week 8 and 16). **d** Edited cells in bone marrow of transplanted mice. GFP is expressed as percentage of CD45+ cells, mean is shown (*$p = 0.012$ AAV vs AAV + RNP, two-tailed Mann–Whitney test). **e** GFP positive cells in HSPCs (CD34), myeloid (CD33), B (CD19) and T (CD3) cells in bone marrow of transplanted mice. Each line represents one animal ($n = 8$). **f** GFP positive cells in HSPCs (CD34) and in a more primitive HSPC subset (CD34+CD38−) in bone marrow of transplanted mice ($n = 4$; ns: $p = 0.8$, two-tailed Mann Whitney test). **g** Western blot on CD34-derived BFU-E from mice engrafted with untreated (UT), transduced (AAV) or KI HSPCs (AAV + RNP). Anti-HA tag and anti-β tubulin antibodies were used ($n = 2$, BFU-E pooled from 2 mice). Source data are in the Source Data file.

Overall, these data show that KI HSPCs can engraft NSG mice and reconstitute all hematopoietic lineages.

## Discussion

We developed an ex vivo platform for efficient gene targeting in human HSCs and robust expression of therapeutic transgenes by the erythroid lineage. By inserting transgenes under the control of the endogenous α-globin gene promoter, we demonstrated that erythroblasts derived from KI HSPCs ex vivo can express and secrete different therapeutic proteins, which retain their enzymatic activity and cross-correct the metabolic defect of patient's cells. In addition, KI HSPCs were able to engraft in vivo and maintained multilineage differentiation potential, we thus expect that our strategy can be used as platform to treat genetic and non-genetic disorders.

We demonstrated that the α-globin locus can be used as a safe harbor for transgene KI in HSCs. In particular, we showed that our selected Cas9/gRNA targeting α-globin 5′ UTR is: (i) efficient in inducing DSB in HSPCs (up to 80%); (ii) safe, as no effect on HSPC multipotency and hemoglobin expression was observed; (iii) specific for α-globin genes, as no predominant off-targets were detected. To further improve the safety profile of this approach, we can envisage the use of Cas9 variants, e.g., high-fidelity[33] or nickase[34]; nonetheless, ad hoc DNA analysis for major chromosomal alterations will be required before moving to clinical testing[35].

Using the described 5′UTR gRNA and an AAV6 vector carrying a promoterless transgene we achieved efficient HDR-based integration in the α-globin locus (above 50%). Although transgene integration will result in knockout of the targeted α-globin allele, this should not be a concern since α-globin genes are redundant and a reduction of 50% of α-globin chain is clinically asymptomatic[14]. In addition, while it is theoretically possible to achieve 4 transgene integrations (1 for each HBA gene), KI efficiency is mostly limited to 1 transgene per cell (Figs. 3d, k, 4h), minimizing the risk of causing α-thalassemia.

Transgene expression was limited to the erythroid lineage and increased following erythroid maturation, as expected from the endogenous α-globin promoter. Importantly, we showed that erythroblasts are able to synthetize and secrete different functional enzymes; secreted LAL was uptaken by patient's fibroblast and correctly sorted to lysosomes to reduce pathological cholesterol accumulation, suggesting that secreted enzymes are properly processed to enter the mannose-6-phosphate pathway[36]. Overall, these results show the versatility of our platform and support its application to other LSD.

By transplanting HSPCs in humanized NSG, we demonstrated that KI HSPCs can repopulate the bone marrow and give rise to progenitors and differentiated hematopoietic lineages. Unfortunately, since NSG and other immunodeficient mouse models do not support significant human erythropoiesis and prevent the in vivo assessment of this erythroid-based platform[32,37], we performed ex vivo erythroid differentiation of bone marrow isolated CD34+ cells confirming that HSCs can still differentiate and express the integrated transgene. Future experiment of KI in mouse HSPCs carrying the human α-globin locus will allow in vivo erythroid differentiation and direct assessment of the steady-state expression levels achievable with our strategy[38].

Protein replacement therapies have proven to be a life-saving therapy for patients affected by rare genetic diseases[1]. However, PRT requires frequent costly injections with a peak-and-trough serum kinetics, which reduce patients' compliance to the therapy and efficacy of treatment[39], and it is affected by development of anti-drugs antibodies, which negatively influence drug bioavailability and activity[40]. Instead, gene therapy can provide constant serum level of therapeutic proteins with a single administration and can induce immune tolerance to the expressed transgene[41]. In particular, the idea of integrating a therapeutic transgene in a safe and highly transcribed genomic locus has been already described for the albumin gene[5–7] and is now under clinical evaluation (NCT03041324, NCT02695160). However, this in vivo approach is hampered by pre-existing liver conditions, pre-existing neutralizing antibodies and cell-mediated immune responses against AAV vectors used to deliver transgenes or nucleases, thus severely limiting the number of potentially treatable patients[8]. To avoid these issues, autologous HSCs can be successfully engineered ex vivo by LV to express transgenes in ubiquitous[42,43] or lineage-restricted manner[44–46], including erythroid lineage[15,16,47,48]; however, the semi-random integration pattern of LV is intrinsically associated with the risk of inactivating an oncosuppressor and transactivating an oncogene. Our strategy promises to be a safer option since transgene integration is targeted to a safe locus, no exogenous promoter is required and transgene expression is truly restricted to erythroid cells, which can induce immune tolerance to exogenous proteins[49–52]. In addition, transgene expression achieved by targeted integration outperformed a LV carrying an erythroid-specific promoter[27,53], which can only partially replicate the complex regulation of a genomic locus due to vector capacity limitations and different chromatin context. The benefit associated to our strategy is twofold: (i) need of fewer modified HSPCs; (ii) higher expression potential.

Our approach still requires bone marrow transplantation of HSPCs, but on-going improvements of HSC mobilization and conditioning regimens will facilitate this procedure[54,55]. In addition, we will explore alternative DNA donor delivery system, e.g. IDLV or non-viral vectors[56], or transient p53 inactivation[29] as means to avoid the negative effect of AAV6 on HSPC engraftment potential[28]. Finally, we will have to assess in vivo if over-expression of transgenes in erythroid precursor cells can have an effect on the HSCs niche in the bone marrow or on erythrocyte differentiation, half-life and clearance[57]. Previous experiments using LV to express different proteins from erythrocytes did not show any impact on erythropoiesis[15–17,58]; however, transgene-specific effects should be carefully evaluated.

Finally, we will engineer transgene sequence with blood-brain barrier shuttle peptides[48,59] to treat LSD with central nervous system involvement, a severe limitation of current PRT[60].

In conclusion, we identified the human α-globin gene as a safe genomic locus for transgene KI in HSCs and erythroid-specific overexpression of therapeutic protein. Future in vivo tests will elucidate the therapeutic potential of this CRISPR-Cas9 based HSC-platform for PRT, especially for LSD and hemophilia.

## Methods

**Plasmids.** Different gRNA protospacers were cloned in hU6-gRNA encoding plasmid (Addgene plasmid # 53188) after BbsI disgestion[61].

Promoter trap encodes for a promoterless GFP reporter (with bovine growth hormone polyA) followed by a puromycin resistance gene under the control of the human phosphoglycerate kinase 1 (PGK) promoter with a SV40 polyA. For intron traps, we added a synthetic intron with splice acceptor site (adapted from[62] and a self-cleaving peptide from porcine teschovirus-1 (P2A)[63] in frame (+0 or +2) at 5′ of GFP cDNA (Supplementary Fig. 2a and Supplementary Methods). These cassettes were inserted in a standard lentiviral vector (LV) backbone[64] in antisense orientation with respect to its LTR.

According to the experiments, GFP and puromycin sequences were exchanged with enzyme cDNA and GFP respectively. F9 (Gene ID: 2158, R338L substitution[26], GLA (Gene ID: 2717), IDUA (Gene ID: 3425) and LAL (Gene ID: 3988) cDNA with a C-terminal 3xHA tag (1xHA: TATCCCTATGACGTGCCT GATTACGCC) and arms of homology (250 bp each) were synthetized by Genscript (Piscataway, NJ) and cloned in a standard AAV vector backbone (AAV2) in sense orientation with respect to its ITR. Upon successful HDR, transgene translation starts from the same ATG as the endogenous α-globin ATG for 5′ UTR integration or after translation of a fragment of α-globin chain for IVS2

integration (α-globin-2A-GFP). Homology arms for 5′UTR α-globin integration are: upstream, chr16:172,642-172,892; downstream, chr16:172,893-173,142 (hg38).

Homology arms for IVS2 α-globin integration are: upstream, chr16:173,135-173,385; downstream, chr16:173,386-173,636 (hg38).

**Vector productions.** LV were produced by transient transfection of 293T using third-generation packaging plasmid pMDLg/p.RRE (or pMDLg/p.RRE.D64V for integrase defective vectors; IDLV), pK.REV, and pseudotyped with the vesicular stomatitis virus glycoprotein G (VSV-G) envelope. LV/IDLV were titrated in HCT116 cells[65] and HIV-1 Gag p24 content was measured by ELISA (Perkin-Elmer) according to manufacturer's instructions.

All recombinant single-stranded AAV2/6 used in this study were produced using an adenovirus-free triple transfection method of HEK293 and purified by two sequential cesium chloride (CsCl) density gradients or by single affinity chromatography (AVB Sepharose; GE Healthcare, Chicago, IL). The final product was formulated in sterile phosphate buffered saline containing 0.001% of pluronic (F68; Sigma Aldrich, Saint Louis, MO), and stored at −80 °C[66,67].

**Cell culture and reagents.** K562 cells (ATCC® CCL-243) were maintained in RPMI 1640 medium containing 2 mM glutamine and supplemented with 10% fetal bovine serum (FBS, Lonza, Basel, Switzerland), 10 mM HEPES, 1 mM sodium pyruvate and penicillin and streptomycin (100U/ml each; Gibco, Waltham, MA, USA). HUDEP-2 cells[18] were maintained in StemSpan SFEM (StemCell Technologies, Vancouver, BC, Canada) supplemented with 2 mM glutamine, 100U/ml penicillin and streptomycin (Gibco, Waltham, MA, USA), Epo 3 U/ml, SCF 50 ng/ml (PeproTech, Rocky Hill, NJ, USA); 1 μg/ml doxycycline and 1 μM dexamethasone (Sigma Aldrich, St. Louis, MO, USA). Cells were differentiated in Iscove's Modified Dulbecco's Medium (IMDM) with 2 mM glutamine, 100U/ml penicillin and streptomycin (Gibco, Waltham, MA, USA Gibco, Waltham, MA, USA), Epo 3 U/ml, SCF 50 ng/ml (PeproTech, Rocky Hill, NJ, USA), 5% AB human serum (Biowest, Nuaillé, France), 10 μg/ml insulin, 330 μg/ml Holo-Transferrin, 2 U/ml heparin, 1 μg/ml doxycycline (Sigma Aldrich, St. Louis, MO, USA). Doxycycline and SCF were removed after 5 days of differentiation[68]. HUDEP-2 single cell clones were obtained by limiting dilution. Human primary fibroblasts were obtained from the NIGMS Human Genetic Cell Repository at the Coriell Institute for Medical Research: GM 11851 A (Wolman disease fibroblasts) and GM 08333 C (healthy donor fibroblasts). Fibroblasts were maintained in DMEM medium supplemented with 2 mM Glutamax (Gibco, Waltham, MA, USA), 15% of FBS (Lonza, Basel, Switzerland) and 100U/ml of penicillin/Streptomycin (Gibco, Waltham, MA, USA). *Streptococcus pyogenes* (Sp)Cas9 protein (with 2 nuclear localization signals, NLS; provided by J.P. Concordet) was expressed in E. coli strain BL21 Rosetta 2. The protein was purified by a combination of affinity, ion exchange and size exclusion chromatographic steps[69]. tracrRNA and crRNA were purchased from Integrated DNA Technologies, resuspended and annealed by manufacturer's instructions. Chemically modified single guide RNA were purchased from Synthego. Primer and probes for PCR were purchased from Sigma-Aldrich (St. Louis, MI, USA) and Integrated DNA Technologies (Coralville, IA, USA).

**Editing experiments in K562.** gRNA screening: To generate a stable clone of Cas9-expressing K562 (K562-Cas9), K562 cells were transduced with a lentiviral vector (Addgene #52962) expressing SpCas9 and a blasticidin resistance cassette, selected with blasticidine (10ug/ml; Sigma-Aldrich, St. Louis, MI, USA) and subcloned. $2.5 \times 10^5$ K562-Cas9 cells were transfected with 200 ng of gRNA-containing plasmid using NucleofectorAmaxa 4D (Lonza, Basel, Switzerland) with SF Cell Line 4D-Nucleofector Kit, K562 program. 48 hours after transfection, cells were harvested and DNA was extracted for molecular analysis.

Targeted integration: $5 \times 10^5$ of K562-Cas9 cells were transduced at multiplicity of infection (MOI) 50 with integrase defective lentiviral vectors (IDLV) containing a promoterless GFP and a constitutively expressed puromycin-resistance gene. After 24 h cells were washed and $2.5 \times 10^5$ of transduced cells were transfected with 200 ng of gRNA-containing vector as previously described. Cells were selected with puromycin (5 ug/ml, Sigma-Aldrich, St. Louis, MI, USA) and sorted for GFP positivity using MoFlocell sorter (Beckman Coulter, Pasadena, CA, USA). Erythroid differentiation was induced with 50 μM Hemin (Sigma-Aldrich, St. Louis, MI, USA) and monitored for 4 days. As K562 differentiation is heterogeneous, to determine differentiation status cells were stained with an anti-Glycophorin A (GYPA) antibody (see list) and GFP expression was analyzed by flow cytometry.

**CD34+ cell culture, transfection and transduction.** Human umbilical cord blood (UCB) samples were provided by Centre Hospitalier Sud-Francilien (CHSF, Evry, France) and processed according to France bioethics laws (declaration DC-2012-1655 to the French Ministry of Higher Education and Research). CD34+ cells were purified by immunomagnetic selection with AUTOMACS PRO (Miltenyi Biotec, Paris, France) after immunostaining with CD34 MicroBead Kit (Miltenyi Biotec, Paris, France). Mobilized peripheral blood (MPB) and UCB CD34+ were also obtained by Cliniscience (Nanterre, France). MPB- or UCB-derived HSPCs were thawed and cultured in prestimulation media for 48 h (StemSpan, Stemregenin-1

0.75uM, StemCell technologies, Vancouver, BC, Canada; rhSCF 300 ng/ml, Flt3-L 300 ng/ml, rhTPO 100 ng/ml and IL-3 20 ng/ml, CellGenix, Freiburg, Germany). Specific crRNA and scaffold tracrRNA were annealed (Integrated DNA Technologies, Coralville, IA, USA) or diluted (Synthego, CA, USA) following manufacturer's instruction and ribonucleoprotein complexes were formed with 30 pmol of spCas9 (ratio 1:1.5). $2.5 \times 10^5$ HSPCs per condition were transfected with RNP complex using P3 Primary Cell 4D-Nucleofector kit (CA137 program). In KI experiments, 15 min after transfection HSPCs were transduced with AAV6 vectors for 6 h (MOI 10000-30000), washed and left in prestimulation media for additional 48 h. Lentiviral transduction of HSPCs was performed in retronectin-coated plates (5ug/cm2) for 6 h in the presence of 4 μg/ml protamine sulfate (Sanofi Aventis, Paris, France)[65]. After manipulation, HSPCs were cultured in erythroid differentiation medium (StemSpan, StemCell Technologies, Vancouver, BC, Canada; SCF 20 ng/ml, Epo 1 u/ml, IL3 5 ng/ml, Dexamethasone 2 μM and Betaestradiol 1 μM; Sigma-Aldrich, St.Louis, MI, USA) or in semisolid Methocult medium (colony-forming cells (CFC) assay, H4435, StemCell Technologies, Vancouver, BC, Canada) for 14 days. Colonies were counted and identified according to morphological criteria (BFU-E, CFU-G/GM, and CFU-GEMM). In some experiments, BFU-E were picked and cultured in erythroid progenitor expansion medium (StemSpan SFEM, StemCell Technologies, Vancouver, BC, Canada; Epo 20 ng/ml, SCF 100 ng/ml, insulin-like growth factor-1 (IGF-1) 50 ng/ml (Pepro-Tech, Rocky Hill, NJ, USA); and 2 μM dexamethasone (Sigma Aldrich, St. Louis, MO, USA) for 3-5 days[70].

**Flow cytometry.** Cells were fixed and/or permeabilized using Cytofix/Cytoperm™ (BD Bioscience, San Jose, CA, USA) according to manufacturer's instructions. For live cells analysis, viability was assessed using Zombie Yellow dye (BioLegend, San Diego, CA, USA) as per manufacturers' instructions to exclude dead cells from the analysis. Negative controls were obtained by staining cells only with the isotype control antibodies. For engraftment studies, an Fc Receptor Binding Inhibitor antibody was used to block unspecific binding of mouse Ab to human cells, as per manufacturers' instructions. Cells were analyzed using CytoFLEX S (Beckman Coulter, Pasadena, CA, USA) or SP6800 Spectral Analyzer (Sony, Tokyo, Japan); data were elaborated with CytExpert (Beckman Coulter, Pasadena, CA, USA) or FlowJo software (Tree Star, OR, USA).

We used MoFlocell sorter (Beckman Coulter, Pasadena, CA, USA) to select for live GFP positive cells. See Supplementary Table 4 for antibodies list and Supplementary methods for gating strategy of human engrafted cells.

**DNA analysis.** Genomic DNA was extracted with QIAamp DNA Micro Kit (Qiagen, Hilden, Germany) or QuickExtract™ DNA Extraction Solution (Lucigen, Middelton, WI, USA).

Quantification of editing efficiency (InDel): 50 ng of genomic DNA were used to amplify the region that spans the cutting site of each gRNA using KAPA2G Fast ReadyMix (Kapa Biosystem, Wilmington, MA, USA). After Sanger sequencing (Genewiz, Takeley, UK), the percentage of Insertions and deletions (InDel) was calculated using TIDE software[22]. See Supplementary Table 3 for primer sequences.

Droplet digital PCR: ddPCR was performed according to manufacturer's instruction using ddPCR Supermix for Probes No dUTP (Biorad, Hercules, CA, US) and 1–50 ng of genomic DNA digested with HindIII (New England Biolabs, Ipswich, MA, USA). Droplets were generated using AutoDG Droplet Generator and analyzed with QX200 droplet reader; data analysis was performed with QuantaSoft (Biorad, Hercules, CA, US).

To quantify HBA2 copy number, primers and probe were designed on the 3′ UTR of HBA2 gene, as it differs significantly from HBA1.

To quantify on-target transgene integration events, primers and probe were designed spanning the donor DNA-genome 3′ junction. Human albumin (ALB) or ZNF843 were used as reference for copy number evaluation (assay ID: dHsaCP2506312, Biorad, Hercules, CA, US). Percentage of on-target integration obtained by ddPCR nicely correlated with GFP values obtained by FACS in KI cells (Supplementary Methods). See Supplementary Table 3 for primer and probe sequences.

**RNA extraction and RT-qPCR.** Total RNA was purified using RNeasy Micro kit (Qiagen, Hilden, Germany) and reverse-transcribed using Transcriptor First Strand cDNA Synthesis Kit (Roche, Basel, Switzerland). qPCR was performed using Maxima Syber Green/Rox (Life Scientific, Thermo-Fisher Scientific, Waltham, MA, US). Primers and probes were optimized using the standard curve method to reach 100% ± 5% efficiency. The relative expression of each target gene was normalized using human GAPDH as a reference gene (NM_002046.6) and represented as $2^{\wedge}\Delta Ct$ for each sample or as fold changes ($2^{\wedge}\Delta\Delta Ct$) relative to the control. See Supplementary Table 3 for primer sequences.

**Protein quantification and Western blot.** FIX detection: FIX antigen in supernatants was measured with an ELISA assay using a standard curve with known amount of human FIX. A microtiter plate is coated with an anti-human FIX antibody (MA1-43012; Thermo-Fisher Scientific, Waltham, MA, US), blocked with PBS-2% bovine serum albumin (BSA) and incubated with diluted supernatants. Protein is detected with a goat anti-human horseradish peroxidase (HRP)-

conjugated antibody (CL20040APHP; Cedarlane, Burlington, Canada)[71]. Samples were analyzed at different dilutions (1/20, 1/40 and 1/100). FIX activity was measured by activated partial thromboplastin time (aPTT)[26]. Protein concentration in diluted supernatant was calculated using a standard curve containing known quantities of hFIX spiked in FIX-deficient plasma.

Western blot: To detect intracellular proteins cells were lysed in RIPA buffer (Sigma-Aldrich, St.Louis, MI, USA) supplemented with protease inhibitor (Roche, Basel, Switzerland), freezed/thawed and centrifuged 10′ at 14,000 at 4 °C. Total protein was quantified using BCA assay (Thermo-Fisher Scientific, Waltham, MA, US). 5–15 μg of protein or 2.5 ul of cell supernatants were denatured at 90 °C for 10′, run under reducing conditions on a 4–12% Bis-tris gel and transferred to a nitrocellulose membrane using iBlot2 system (Invitrogen, Waltham, MA, US). After Ponceau staining (Invitrogen, Waltham, MA, US) membranes were blocked for 2 h with Odyssey blocking buffer (Odyssey Blocking buffer (PBS), Li-Cor Biosciences, Lincoln, NE, USA) and incubated for 1 h with primary antibodies followed by specific secondary antibodies in PBS:Blocking buffer (see Supplementary Table 5 for antibodies list). β-Tubulin was used as loading control. Blots were imaged at 169 μm with Odyssey imager and analyzed with ImageStudio Lite software (Li-Cor Biosciences, Lincoln, NE, USA). After image background subtraction (average method, top/bottom), band intensities were quantified and normalized with tubulin signal. Antibody concentrations, suppliers and catalog numbers are provided in Supplementary Table 5.

LAL activity assay: Protein activity was detected in supernatants as previously described[72,73] with some modifications. Briefly, samples were incubated 10 min at 37 °C with 42 μM Lalistat-2 (Sigma-Aldrich, St. Louis, MI, USA), a specific competitive inhibitor of LAL, or water. Samples were then transferred to a Optiplate 96 F plate (PerkinElmer) where fluorimetric reactions were initiated with 75 μl of substrate buffer (340 μM 4-MUP, 0.9% Triton X-100 and 220 μM cardiolipin in 135 mM acetate buffer pH 4.0). After 10 min, fluorescence was recorded (35 cycles, 30″ intervals, 37 °C) using SPARK TECAN Reader (Tecan, Austria). Kinetic parameters (average rate) were calculated using Magellan Software. LAL activity over untreated samples was quantified using this formula:

$$\frac{\text{Edited sample (without Lalistat − with Lalistat)}}{\text{Untreated sample (without Lalistat − with Lalistat)}}$$

LAL uptake assay: Equal amounts of conditioned medium from KI or control HSPCs during erythroid differentiation were collected, concentrated using Amicon® 10 kDa (Merck, Kenilworth, NJ, USA), diluted with opti-MEM to their original volume (Gibco, Waltham, MA, USA) and filtrated with a 0.22 μm filter (Millipore, Burlington, MA, USA). Processed medium were added to $4.5 \times 10^5$ WD fibroblasts in a 6-well plate. After 3 days, fibroblasts were harvested and pellets were frozen for LAL and cholesterol quantification.

Cholesterol quantification: Total cholesterol was measured with Amplex red (Thermo-Fisher Scientific, Waltham, MA, US) as per manufacturer's instructions. Fluorescence (endpoint) was recorded with SPARK TECAN Reader (Tecan, Austria) and cholesterol content was quantified with a standard curve.

Nile red staining: $4 \times 10^4$ fibroblasts were seeded in a 8⁻well LAB_TEK coverglass (Nunc, Rochester, NY, USA) and cultured in conditioned medium of KI or UT HSPCs. After 3 days, cell were stained with Nile Red (Nile Red staining kit, Abcam, Cambridge, UK) as per manufacturer's instructions. 8 fields for each condition were randomly acquired with an inverted fluorescence microscope (10x magnification; EVOS imaging system, Thermo-Fisher Scientific, Waltham, MA, US) and average fluorescence intensity per cell was calculated with ImageJ[74] using a custom made macro.

**HPLC analysis of globin chains and tetramers**. HPLC analysis was performed using a NexeraX2 SIL-30AC chromatograph (Shimadzu, Kyoto, Japan) and analyzed with LC Solution software. HSPC derived erythroblasts were lysed in water and globin chains were separated using a 250 × 4.6 mm, 3.6 μm Aeris Widepore column (Phenomenex, Torrance, CA, USA). Samples were eluted with a gradient mixture of solution A (water/acetonitrile/trifluoroacetic acid, 95:5:0.1) and solution B (water/acetonitrile/trifluoroacetic acid, 5:95:0.1), monitoring absorbance at 220 nm. Hemoglobin tetramers were separated by HPLC using a cation-exchange column (PolyCAT A, PolyLC, Columbia, MD, USA). Samples were eluted with a gradient mixture of solution A (20 mM bis Tris, 2 mM KCN, pH 6.5) and solution B (20 mM bis Tris, 2 mM KCN, 250 mM NaCl, pH 6.8). The absorbance was measured at 415 nm.

**In vivo experiments**. NOD.Cg-Prkdc^scid^Il2rg^tm1Wjl^/SzJ (NSG) mice were purchased from The Jackson Laboratory (strain 005557) and maintained in specific-pathogen-free (SPF) conditions. This study was approved by ethical committee CEEA-51 and conducted according to French and European legislation on animal experimentation (APAFiS#16499-2018071809263257_v4).

48 h after editing, $5–7 \times 10^5$ CD34 + cells were injected intravenously into female NSG mice after sublethal irradiation (150 cGy). Human cell engraftment and KI levels were monitored at different time points in peripheral blood by flow cytometry using anti human CD45 and HLA-ABC antibodies (see Supplementary Table 4). 16 weeks after transplantation, blood, bone marrow and spleen were harvested and analyzed. Peripheral blood was directly stained and red blood cells lysed during sample fixation (VersaLyse Lysing Solution and IOTest3 Fixative solution, Beckman Coulter, Pasadena, CA, USA).

**Cell purification and enrichment**. Human CD34$^+$ cells were purified from mouse bone marrow by immunomagnetic selection with CD34 MicroBead Kit UltraPure in combination with AUTOMACS PRO (PosselD2 separation program; Miltenyi Biotec, Paris, France). Human CD45 cells from mouse peripheral blood or bone marrow were enriched with CD45 MicroBeads (Possel separation program; Miltenyi Biotec, Paris, France).

**Off-target analysis**. Off-target candidates were predicted in silico using two different software with the following parameters: up to 4 mismatches and no bulges (CRISPOR)[20]; up to 2 mismatches, 1 insertion and 1 deletion tolerated (COSMID)[21].

K562-Cas9 cells were edited at saturation with multiple rounds of transfection with different gRNA; genomic DNA was amplified at predicted off-target sites, Sanger-sequenced and analyzed with TIDE software.

IDLV capture[24] was used for experimental identification of potential off-target sites. K562 cells were transduced with an IDLV expressing a GFP reporter (MOI 100) and subsequently transfected with 30 pmol of Cas9:gRNA complex (1:2). Two weeks later at least $5 \times 10^4$ GFP positive cells were sorted using MoFlocell sorter (Beckman Coulter, Pasadena, CA, USA) and expanded for genomic DNA extraction. LTR vector-genome junctions were amplified by ligation mediated (LM)-PCR as previously described[25]. Briefly, 1 μg genomic DNA was fragmented with Tru91 restriction enzyme (Roche, Basel, Switzerland) and ligated to a TA-protruding double-stranded DNA linker. After SacI digestion (Roche, Basel, Switzerland), multiple nested PCRs were performed with specific primers annealing to the linker and vector LTR. Amplicons ranging from 200 to 500 kb were purified by NucleoSpin Gel and PCR Clean-up kit (Macherey-Nagel, Düren, Germany). 1 μg of the final libraries was subsequently processed with MiSeq Reagent Kit v3 (2 × 300-bp pair-end sequencing) and sequenced to saturation on Illumina MiSeq System (IGA Technology Services, Udine, Italy). Raw reads were processed as previously described, alignments with best scores were kept, and integration sites were identified. IDLV integration sites that mapped within a ±300 bp window were identified as clustered integration sites (CLIS).

**Statistical analyses**. Statistical analyses were performed using GraphPad Prism version 6.00 for Windows (GraphPad Software, La Jolla, CA, USA, "www. graphpad.com"). One-way or two-way analysis of variance (ANOVA) with Tukey's or Dunnet's multiple comparison post-test for three or more groups was performed as indicated (alpha = 0.05). Values are expressed as mean ± standard deviation (SD) as otherwise indicated.

**Reporting summary**. Further information on research design is available in the Nature Research Reporting Summary linked to this article.

## Data availability

The authors declare that all data supporting the findings of this study are available within the paper and Supplementary Information. The IDLV capture data that support the findings of this study have been deposited in NCBI Gene Expression Omnibus (GEO) under accession number GSE133861. [https://www.ncbi.nlm.nih.gov/geo/query/acc.cgi? acc=GSE133861]. Any other relevant data are available upon reasonable request. Source data are provided with this paper.

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

## Acknowledgements

We thank Sabine Charier for advices on NGS engraftment, Anna De Cian, Fatima Amor, Anne Chalumeau and Cecile Loubière for technical help. We also thank Samia Martin and Genethon "Vector Core Facility" for vector production, Jeremy Cosette and Genethon "Imaging and Cytometry Core Facility" for image analysis and FACS sorting, Laetitia van Wittenberghe for help with mice experimentation. We gratefully acknowledge the Conseil Général de l'Essonne (ASTRES) and Genopole Research in Evry for financial help for the purchase of equipment. We thank Anne Galy, Giuseppe Ronzitti, Fulvio Mavilio and the whole Amendola's laboratory for fruitful discussion. We thank Ryo Kurita and Yukio Nakamura for proving HUDEP-2 cells under an MTA with Genethon and Feng Zhang and Charles Gersbach for providing plasmids through Addgene. We are grateful to consenting mothers and to Dr Rigonnot and staff of the Maternity at the Centre Hospitalier Sud-Francilien (CHSF; Evry, France) for providing umbilical cord blood samples. This work was supported by grants to G.P., Bayer (Hemophilia Awards Program), and M.A., (AFM-Telethon, Inserm and Genopole (Chaire Fondagen)). G.P. was supported by the European Union's Horizon 2020 (SCIDNET No 666908). A.S. was partially supported by a PhD fellowship from the French Minister of Higher Education, Research and Innovation via University of Evry.

## Author contributions

G.P. conceived the study, designed and performed experiments, analyzed data and wrote the manuscript. M.L. performed experiments and analyzed data for LAL. E.K., A.F. and A.S. performed experiments. G.C. performed bioinformatics analysis of Cas9/gRNA specificity. P.L. performed aPTT assay for FIX experiments. J.P.C. provided purified SpCas9 protein. M.T. produced and titrated AAV and LV. A.M. performed and analyzed HPLC. M.A. conceived the study, designed experiments, analyzed data and wrote the manuscript.

## Competing interests

G.P. and M.A. are the inventors of a patent describing this HSC-based gene therapy platform (Genetically engineered hematopoietic stem cell as a platform for systemic protein expression; EP18305026.9). The remaining authors declare no competing interests.
