## [Peer Review File · Nature Communications]

Reviewers' Comments:

Reviewer #1:

Remarks to the Author:

In this manuscript by G. Pavani et al., Authors performed targeted integration of a promoter less transgene cassette into the hemoglobin A genes (HBA) of hematopoietic stem/progenitor cells (HSPC) in order to develop a robust protein delivery system based on stem cell-derived erythroid cells. This strategy was validated by showing increasing expression of different transgenes by the HBA promoter during *in vitro* erythroid differentiation of edited human cell lines and cord blood derived HSPC. Then, the Authors provided proof of concept for therapeutic application of this platform by showing secretion and preservation of enzymatic activity of a hyperactive factor IX (F9) and the lysosomal acid lipase (LAL), thus suggesting possible implementation as new treatment for hemophilia or lysosomal storage disorders.

The reported experiments were accurately performed and the results appear convincing as they are often based on multiple lines of independent evidence. Despite, as fairly stated by the Authors, the use of erythroid cells as systemic source for transgene secretion was already reported by means of tissue-specific gene replacement vectors, the concept of exploiting the high transcriptional activity of HBA during erythroid differentiation is new and clever. Indeed, this strategy will allow to maximize transgene overexpression while avoiding the need for multiple vector integrations or strong artificial enhancer/promoters.

Thus, the overall findings of this study can be considered of sufficient novelty and interest for the gene therapy field. However, the following points need to be addressed in order to improve the quality and general interest of this manuscript.

Major points:

Since NHEJ is the most common gene editing readout in HSPC, the choice of targeting a non-translated region of the HBA genes will allow preserving their expression in a good fraction of the cells that does not contain homologous mediated integration. To verify this, the Authors performed quantitative PCR analyses on the HBA genes and found only low levels of genomic deletions between the two consecutive genes and showed a representative HPLC analysis that confirmed presence of the globin A subunit and HbA tetramer. However, since this is an important aspect of the proposed strategy, more experimental detail and scientific discussion have to be provided:

- Genomic inversions that does not delete the HBA2 gene can occur when using a nuclease that cut two targets on the same chromosome. Since this event will result in functional KO of both the HBA genes involved, proper quantification of this rearrangement has to be reported (e.g. by ddPCR on the genomic junctions)

- How many samples were analyzed by HPLC? Which is the fraction of edited cells in the analyzed population?

- Targeted integration of the transgene expression cassette is designed to abrogate the expression of the edited allele. Despite this is not a concern, thanks to the genomic redundancy and the expected limited contribution of KI cells to the reconstituted hematopoiesis after transplant, these points have to be mentioned and discussed in the manuscript.

- When performing homology driven integration into the HBA genes, up of 4 copies of the donor template could be knocked-in due to the high similarity of the homology sequences between the two loci (HBA1 and HBA2). The Authors should quantify these recombination events or comment on why they are not detected (e.g., in Fig.3d how are the integration measured?).

The Authors reported a comprehensive off-target analysis for their selected gRNA, by combining *in silico* prediction and unbiased genome wide assays. However, better description of the results has to be reported to facilitate their correct interpretations:

- Description of Suppl. Table 2 is currently unclear. The Authors claim in the text that "no indels were observed at any of the predicted off-target sites", but from the table it rather appears that there is no difference in the levels of indels detected on HAB15 and control AAVS1 gRNAs. Is this the right interpretation of the table? What about OT16 that shows 2% of indels?

- In Supl Fig.3b, it is not clear from the table why the Authors claim that no unique CLIS for 5'UTR (19 reported in the figure) were identified after correction for random integration (6 reported).

Does this mean that more than one CLIS was found at the on-target site?

- In the methods section it is reported that the levels of indels at the identified off-target sites was measured by TIDE analysis, which has a sensitivity of ~1%. Despite this threshold could be considered sufficient for a proof of concept study, this information has to be clearly stated in the main text to better guide future follow-up studies.

Data on F9 and LAL secretion from the edited cells are an important proof of principle that thereported system allow production of proteins with therapeutic relevance and that at least these two enzymes are not toxic when overexpressed in differentiating HSPC. However, better description of the results and additional discussion points are to be included in the manuscript:

- In Fig.3h, F9 activity should be indicated as aPPT and the scale line is probably mislabeled (are the reported data expressed as fold over untreated?). Data must also be reported as time (seconds) comparing edited and untreated cells.

- Proteins ectopically expressed in differentiating erythrocytes would possibly contain complex post-translational modification, but since no direct and extended experimental proof are provided in the text (only indirect evidence of Glu6P could be argued), this claim have to be attenuated.

- The Authors state that protein replacement therapy has several problems that could be overcome by their proposed strategy. However, the discussion should better address which are these advantages (e.g. single administration, possible induction of tolerance, ...) and which could be possible disadvantage of the new proposed strategy. In particular, it will be important to mention possible problems related to dose control of the therapeutic enzyme after HSC transplantation (e.g. F9 Padua: low expression would result in limited efficacy while high engraftment could generate thrombophilia).

- It could be of interest to add some speculations about possible effects of local over-concentration of some enzymes in the bone marrow, during erythrocytes differentiation, or in the erythrocyte degradation organ after red cell turnover.

Minor points:

- Is it not clear which of the gRNA tested in Fig.1a and Suppl. Fig.1 is used all over the manuscript (HBA15).

- The data reported in Suppl. Fig 4b are not an indication of HSPC multipotency but rather of preservation of their clonogenic differentiation capacity.

- Suppl. Fig. 3c is missing.

Reviewer #2:

Remarks to the Author:

In the article entitled "Ex vivo editing of human hematopoietic stem cells for erythroid expression of therapeutic proteins", Pavani et al., take advantage of robust expression of the HBA loci encoding alpha-globin to develop a system for robust expression of transgenes specifically in erythrocytes. The authors take advantage of an evolutionary HBA tandem duplication (HBA2, HBA1 genes) and tolerance of erythroid function to alpha-globin deficiency, even with the loss of three gene copies. The authors use this approach to express various proteins, which show potential for therapeutic activity in vitro and in vivo.

Although the strategy to achieve reliable tissue-specific transgene expression through gene targeting has already been shown for other loci (such as albumin), reducing novelty of the overall approach, this report appears to be the first example of using HBA as an integration site.

Moreover, erythrocytes derived from primary hematopoietic stem cells, or HSCs themselves, are good candidates for cellular therapy, and the possibility of ex vivo manipulation can improve safety by pre-screening.

While the method has promise, the article in its current form fails to clearly demonstrate the aspect of safety (mentioned multiple times by the authors), nor does it establish a convincing standard for reproduction as a method. Specifically, the gRNAs, targeting vectors, and genotyping assays are poorly described or referenced in the main text. Fundamentally, the choice to target duplicated HBA2/1 sequences suggests the possibility of large deletions or rearrangements which

are not explored with sufficiently with standard molecular assays.

Specific Points:

The authors select the 5'UTR gRNA as the standard for targeting, based on efficiency and expression. However, looking at UCSC Genome Data, this gRNA has perfect homology to both HBA2 and 1. It is well known that cleaving two adjacent loci with nucleases can result in deletions or inversions. The authors state that "All gRNA except 74 target both HBA1 and HBA2." This suggests that deletions or rearrangements would be a prevalent outcome, yet they make little effort (beyond ddPCR) to screen for it. This is necessary data to validate the safety and usability of their system. To be convincing, genomic structural analysis is needed, such as long-range PCR, Southern blot, sequencing, or ddPCR with multiple probes.

Along these lines, the ddPCR analysis of copy number (Fig. S1f) seems to show only a loss of one copy. Moreover, the data in Figure 3b is used to show correlation with GFP+ cells. However, HBA genes are both duplicated and autosomal (4 copies) what is the expected outcome of hetero and homozygous targeting? The authors completely ignore this possibility. Please clarify, and provide supporting data.

The authors use GFP to enrich cells with vector integrations. What is the overall frequency of integration (ie. %GFP+). It seems the data is only shown for Factor IX integration. (Fig 3b Pre Sort). Please include this data to demonstrate the overall targeting efficiency.

In the long run, is GFP a viable solution for therapeutic application? What is a reasonable alternative? Please discuss.

The authors screen multiple gRNAs targeting the 5'UTR or introns, and select two sites as leads. However, the overall lack of detail on the homology arms, gRNAs and genotyping assays (apart from supplemental data) makes it difficult to understand the exact integration site chosen, or the rationale. Moreover, the IVS2 integrations site and trap vector should produce a truncated α -globin protein, which is not screened for (eg. Western blot) or even discussed by the authors.

There are many instances of "data not shown", against Nature policy. Also, colony screening data is not shown for Figure 3k, although HDR rates are reported (7/7).

The screening process for gRNAs is not clear. In Figure 1a – does each grey bar represent a different gRNA? A map of the locus (found in Fig. S1a) or appropriate labelling is required in the main figure.

Throughout the paper, the gRNA used for each experiment is not clear. Which guides were chosen from Supp2e/f to produce the data in Panel g? (HBB targeting) Moreover, the details of the target site are not clear. While the 5'UTR gRNA cuts upstream of the ATG, does the targeting vector correct the position of the insertion to be in-frame with the endogenous ATG, or does it insert directly at the CRISPR/Cas9 cut site?

Genotyping data for insertion at HBB is not shown.

"in line with TIDE-based indels analysis (8.5%±4.9, n=5)"
The TIDE data is not shown?

Remarkably, 5'UTR and IVS2 gRNA did not modify α -globin expression, measured as ratio between α - and β -like globin chains (Figure 1e and supplementary figure 1d). Was modification expected? Did the authors make an effort to map or avoid regulatory elements? What was the nature of the indels formed? The sequence analysis of indels is not made available.

In the first set of results, the authors state "without affecting HSPC viability, differentiation potential and hemoglobin expression, thus representing ideal genomic sites for KI.", However these data do not yet demonstrate transgene expression, and therefore do not support ideal sites for KI. Please re-write.

What are the predicted PCR fragment sizes in S2c/d? Presumably the PCR fragments differ in sizes amongst the clones because of NHEJ insertion of the IDLV. Please explain.

Do the two bars in Fig S1b represent two independent experiments, or two different gRNAs?

Lanes are not labelled in Fig S2i, and there is no schematic for the PCR assay.

"KI occurred mostly through HDR (>90% of CFC, n=40; supplementary fig 2j)."
Fig S2j shows one sequencing result, not a summary of targeting efficiencies.

The terms BFU-E, CFU-E, and CFU-GEMM are used in the text and need to be defined there. Is CFU-GM in Fig S2i the same as CFU-GEMM? The authors must be consistent.

"PCR analysis of individual CFC showed integration in both red and white colonies (Supplementary figure 2i)." What is meant by red and white?

Figure 2h, please use scale bars, not magnification. Also, why are only positive cells shown? Including examples of GFP negative cells would be more convincing.

"no indels were observed at any of the predicted off-target sites (Supplementary table 2)."
If that is the case, then what is the data in the table (column 4, last two columns)?

"Off-target candidates were predicted in silico using two different software"
Please state the software or provide a URL.

"BFU-E derived erythroblasts were capable of secreting FIX (supplementary figure 3c)."
Should the comment cite Fig. S4d?

Point-by-Point reply to Reviewers' comment

Revision manuscript NCOMMS-19-40192-T

We would like to thank the reviewers for their critical comments, which helped us to improve the current version of the manuscript by Pavani et al. entitled "*Ex vivo editing of human hematopoietic stem cells for erythroid expression of therapeutic proteins*".

Please find below a detailed point-by-point reply (in red) to each comment.

Referee #1:

"In this manuscript by G. Pavani et al., Authors performed targeted integration of a promoter less transgene cassette into the hemoglobin A genes (HBA) of hematopoietic stem/progenitor cells (HSPC) in order to develop a robust protein delivery system based on stem cell-derived erythroid cells. This strategy was validated by showing increasing expression of different transgenes by the HBA promoter during in vitro erythroid differentiation of edited human cell lines and cord blood derived HSPC. Then, the Authors provided proof of concept for therapeutic application of this platform by showing secretion and preservation of enzymatic activity of a hyperactive factor IX (F9) and the lysosomal acid lipase (LAL), thus suggesting possible implementation as new treatment for hemophilia or lysosomal storage disorders. The reported experiments were accurately performed and the results appear convincing as they are often based on multiple lines of independent evidence. Despite, as fairly stated by the Authors, the use of erythroid cells as systemic source for transgene secretion was already reported by means of tissue-specific gene replacement vectors, the concept of exploiting the high transcriptional activity of HBA during erythroid differentiation is new and clever. Indeed, this strategy will allow to maximize transgene overexpression while avoiding the need for multiple vector integrations or strong artificial enhancer/promoters.

Thus, the overall findings of this study can be considered of sufficient novelty and interest for the gene therapy field. However, the following points need to be addressed in order to improve the quality and general interest of this manuscript.

Major points:

Since NHEJ is the most common gene editing readout in HSPC, the choice of targeting a non-translated region of the HBA genes will allow preserving their expression in a good fraction of the cells that does not contain homologous mediated integration. To verify this, the Authors performed quantitative PCR analyses on the HBA genes and found only low levels of genomic deletions between the two consecutive

genes and showed a representative HPLC analysis that confirmed presence of the globin A subunit and HbA tetramer. However, since this is an important aspect of the proposed strategy, more experimental detail and scientific discussion have to be provided:

-Genomic inversions that does not delete the HBA2 gene can occur when using a nuclease that cut two targets on the same chromosome. Since this event will result in functional KO of both the HBA genes involved, proper quantification of this rearrangement has to be reported (e.g. by ddPCR on the genomic junctions)”

We thank the referee for the useful comment.

In this revised manuscript, we **better characterized the outcome of CRISPR/Cas9 genomic modification by:**

- i) **systematic quantification of InDels** in edited HSPCs (figure 1 C already present; **new Supplementary figure 4 B and Supplementary figure 5A)**
- ii) **systematic quantification of transgene integration by on target ddPCR** in edited HSPCs (**new Supplementary figure 4C and Supplementary figure 5B)**
- iii) **systematic quantification of HBA2 deletions** in edited HSPCs (Supplementary figure 1F, already present; **new Supplementary figure 4D and Supplementary figure 5C)**
- iv) **genotyping 69 BFU-E** for HBA2 deletion and transgene integration (**new Supplementary figure 5E-F)**
- v) **analyzing 28 HUDEP-2 single cell clones to quantify HBA1 vs HBA2 targeted integration** (**new Supplementary figure 4A)**

We have added this information in the results and materials and methods sections of the revised manuscript.

We also tried to assess HBA2 inversion in bulk population of HSPCs edited with HBA15 RNP using 6 different PCR and we could not detect any specific amplification.

This is in line with a recent publication by Li et al. for the human γ -globin locus, which is similar to the α -globin one for the presence of evolutionary duplicated genes (HGB1/HBG2 vs HBA1/HBA2). Specifically, the authors performed CRISPR/Cas9 editing of the γ -globin locus in HSPCs and they could detect genomic deletion but not genomic inversion. (PMID: 29789357; Fig 7).

Although we are persuaded that the lack of genomic inversion in our edited HSPCs is a true data, we decided not to insert this information in the revised version of the manuscript since we could not validate our PCR primers using an artificial template as positive control.

The reason is twofold:

- 1) Two manufacturers independently confirmed that the presence of repetitive sequences and the high GC content of the α -globin locus (PMID: 11157800) makes the DNA synthesis of such template not possible with current technology.
- 2) The forced lockdown of the lab for the current coronavirus pandemic does not allow us to perform any additional attempt for an unknown amount of time (probably 2 months).

If necessary, we can explain in the manuscript that we did not detect any HBA2 inversion by PCR analysis, although we cannot formally exclude a false negative result.

“How many samples were analyzed by HPLC? Which is the fraction of edited cells in the analyzed population?”

We added these details in the revised version of the manuscript (legend Figure 1 F-G).

“Targeted integration of the transgene expression cassette is designed to abrogate the expression of the edited allele. Despite this is not a concern, thanks to the genomic redundancy and the expected limited contribution of KI cells to the reconstituted hematopoiesis after transplant, these points have to be mentioned and discussed in the manuscript.”

We thank the referee for the useful comment. **We have now better explained this point in the conclusion paragraph** of the revised version of the manuscript.

“Although transgene integration will result in knockout of the targeted α -globin allele, this should not be a concern since α -globin genes are redundant and a reduction of 50% of α -globin chain is clinically asymptomatic¹⁴. In addition, although it is theoretically possible to achieve 4 transgene integration (1 for each HBA gene), KI efficiency is mostly limited to 1 transgene per cell (Figure 3D and K, Figure 4H), minimizing the risk of causing α -thalassemia.”

“When performing homology driven integration into the HBA genes, up of 4 copies of the donor template could be knocked-in due to the high similarity of the homology sequences between the two loci (HBA1

and HBA2). The Authors should quantify these recombination events or comment on why they are not detected”

We thank the referee for the useful comment. We quantified the number of donor template targeted integration and **we observed mostly 1 integration per cell, less frequently 2 integration (Figure 3D and K, Figure 4H). We never observed 3 or 4 integrations**, which are either absent or too rare to be detected with our quantification methods (ddPCR).

As suggested, **we comment this aspect in the conclusion paragraph** of the revised version of the manuscript (as above):

“Although transgene integration will result in knockout of the targeted α -globin allele, this should not be a concern since α -globin genes are redundant and a reduction of 50% of α -globin chain is clinically asymptomatic²⁴. In addition, although it is theoretically possible to achieve 4 transgene integration (1 for each HBA gene), KI efficiency is mostly limited to 1 transgene per cell (Figure 3D and K, Figure 4H), minimizing the risk of causing α -thalassemia.”

“(e.g., in Fig.3d how are the integration measured?).”

In Figure 3D the number of targeted integration events was measured **in each cell clone by ddPCR for on-target integration (primer specific for the AAV-genome junction). This information has been added** in the revised version of the manuscript (**legend Figure 3D**).

“The Authors reported a comprehensive off-target analysis for their selected gRNA, by combining in silico prediction and unbiased genome wide assays. However, better description of the results has to be reported to facilitate their correct interpretations:

Description of Suppl. Table 2 is currently unclear. The Authors claim in the text that “no indels were observed at any of the predicted off-target sites”, but from the table it rather appears that there is no difference in the levels of indels detected on HAB15 and control AAVS1 gRNAs. Is this the right interpretation of the table?”

“What about OT16 that shows 2% of indels?”

We thank the referee for the useful comment. We could not detect any difference in the levels of indels detected between HAB15 and control AAVS1 gRNAs; therefore, **we rewrote the result paragraph accordingly.**

The **TIDE software**, which was used to analyze the Sanger sequencing results of the PCR for the predicted off-targets, **is not reliable below ~2%** (PMID: 25300484); therefore, OT16 off-target was considered negative with a 2% threshold. In addition, OT16 is in an intron of a gene minimally expressed in blood (<https://www.gtexportal.org/home/gene/ENSG00000167394>); therefore, no additional analyses were performed.

Both these information have been added in result section of the revised version of the manuscript. *“Although on-target activity reached >85%, we could not detect any difference between HBA15 and control AAVS1 gRNA at any of the predicted HBA15 off-target sites (Supplementary Table 2) (with a technical threshold of >2% of TIDE software).”*

“In Supl Fig.3b, it is not clear from the table why the Authors claim that no unique CLIS for 5’UTR (19 reported in the figure) were identified after correction for random integration (6 reported). Does this mean that more than one CLIS was found at the on-target site?”

We thank the referee for the useful comment. The number of CLIS indicated in the table (n CLIS) for each gRNA was the sum of all the CLIS found in each replicate (if the same CLIS was found in 3 replicates we would write n CLIS = 3). We agree that this can be misleading; therefore, **in the revised version of the manuscript we added two new columns to the Supplementary Figure 3C table:**

- 1) CLIS unique: indicating the number of unique CLIS for each gRNA (9 for HBA15)**
- 2) gRNA specific CLIS: indicating the number of CLIS that are specific for each gRNA, removing the CLIS that are in common among different gRNA or IDLV control.**

“In the methods section it is reported that the levels of indels at the identified off-target sites was measured by TIDE analysis, which has a sensitivity of ~1%. Despite this threshold could be considered sufficient for a proof of concept study, this information has to be clearly stated in the main text to better guide future follow-up studies.”

We thank the referee for the useful comment. **We now modified the result paragraph** (as above): *“Although on-target activity reached >85%, we could not detect any difference between HBA15 and control AAVS1 gRNA at any of the predicted HBA15 off-target sites (Supplementary Table 2) (with a technical threshold of >2% of TIDE software).”*

“Data on F9 and LAL secretion from the edited cells are an important proof of principle that the reported system allow production of proteins with therapeutic relevance and that at least these two enzymes are not toxic when overexpressed in differentiating HSPC. However, better description of the results and additional discussion points are to be included in the manuscript:

In Fig.3h, F9 activity should be indicated as aPTT and the scale line is probably mislabeled (are the reported data expressed as fold over untreated?). Data must also be reported as time (seconds) comparing edited and untreated cells.”

We thank the referee for the useful comment and we now added the Supplementary Figure 4E with the seconds of the aPTT test.

In the materials and methods paragraph, we explain better how FIX was quantified.

We corrected the Y axis of Fig 3I.

“Proteins ectopically expressed in differentiating erythrocytes would possibly contain complex post-translational modification, but since no direct and extended experimental proof are provided in the text (only indirect evidence of Glu6P could be argued), this claim have to be attenuated.”

We thank the referee for the useful comment. We have now rephrased this claim: “Importantly, erythroblasts are able to synthesize and secrete different functional enzymes”

“The Authors state that protein replacement therapy has several problems that could be overcome by their proposed strategy. However, the discussion should better address which are these advantages (e.g. single administration, possible induction of tolerance, ...) and which could be possible disadvantage of the new proposed strategy. In particular, it will be important to mention possible problems related to dose control of the therapeutic enzyme after HSC transplantation (e.g. F9 Padua: low expression would result in limited efficacy while high engraftment could generate thrombophilia). - It could be of interest to add some speculations about possible effects of local over-concentration of some enzymes in the bone marrow, during erythrocytes differentiation, or in the erythrocyte degradation organ after red cell turnover.”

We thank the referee for the useful comment. This was previously not possible due to the word count limit. As the Editor has relaxed this limit, we now extended the discussion paragraph as suggested.

“Protein replacement therapies have proven to be a life-saving therapy for patients affected by rare genetic diseases¹. However, PRT requires frequent costly injections with a peak-and-trough serum kinetics, which

affect patients' compliance to the therapy and efficacy of treatment³⁹, and it is affected by development of anti-drugs antibodies, which negatively influence drug bioavailability and activity⁴⁰. Instead, gene therapy can provide constant serum level of therapeutic proteins with a single administration and can induce immune tolerance to the expressed transgene⁴¹."

"Finally, we will have to assess in vivo if over-expression of transgenes in erythroid precursor cells can have an effect on the HSCs niche in the bone marrow, on erythrocyte differentiation, half-life and clearance⁵⁷. Previous experiments using LV to express different proteins from erythrocytes did not show any impact on erythropoiesis^{15-17, 58}; however, transgene-specific effects should be carefully evaluated."

"Minor points:

Is it not clear which of the gRNA tested in Fig.1a and Suppl. Fig.1 is used all over the manuscript (HBA15)."

We thank the referee for the useful comment. We have now color-coded the gRNA in Figure 1 and in Supplementary Figure 2.

"The data reported in Suppl. Fig 4b are not an indication of HSPC multipotency but rather of preservation of their clonogenic differentiation capacity."

We thank the referee for the useful comment. We have now corrected the sentence as suggested:
"...did not affect HPSC clonogenic differentiation capacity (Supplementary figure 4g)"

"Suppl. Fig. 3c is missing."

We thank the referee for the useful comment. We have now corrected the mistake.

Referee #2:

"In the article entitled "Ex vivo editing of human hematopoietic stem cells for erythroid expression of therapeutic proteins", Pavani et al., take advantage of robust expression of the HBA loci encoding alpha-globin to develop a system for robust expression of transgenes specifically in erythrocytes. The authors take advantage of an evolutionary HBA tandem duplication (HBA2, HBA1 genes) and tolerance of erythroid function to alpha-globin deficiency, even with the loss of three gene copies. The authors use this approach to express various proteins, which show potential for therapeutic activity in vitro and in vivo.

Although the strategy to achieve reliable tissue-specific transgene expression through gene targeting has already been shown for other loci (such as albumin), reducing novelty of the overall approach, this report appears to be the first example of using HBA as an integration site. Moreover, erythrocytes derived from primary hematopoietic stem cells, or HSCs themselves, are good candidates for cellular therapy, and the possibility of ex vivo manipulation can improve safety by pre-screening.

While the method has promise, the article in its current form fails to clearly demonstrate the aspect of safety (mentioned multiple times by the authors), nor does it establish a convincing standard for reproduction as a method. Specifically, the gRNAs, targeting vectors, and genotyping assays are poorly described or referenced in the main text. Fundamentally, the choice to target duplicated HBA2/1 sequences suggests the possibility of large deletions or rearrangements which are not explored with sufficiently with standard molecular assays.

Need to ameliorate this

Specific Points:

The authors select the 5'UTR gRNA as the standard for targeting, based on efficiency and expression. However, looking at UCSC Genome Data, this gRNA has perfect homology to both HBA2 and 1. It is well known that cleaving two adjacent loci with nucleases can result in deletions or inversions. The authors state that "All gRNA except 74 target both HBA1 and HBA2." This suggests that deletions or rearrangements would be a prevalent outcome, yet they make little effort (beyond ddPCR) to screen for it. This is necessary data to validate the safety and usability of their system. To be convincing, genomic structural analysis is needed, such as long-range PCR, Southern blot, sequencing, or ddPCR with multiple probes."

We thank the referee for the useful comment.

In this revised manuscript, we **better characterized the outcome of CRISPR/Cas9 genomic modification by:**

- vi) **systematic quantification of InDels** in edited HSPCs (figure 1 C already present; **new Supplementary figure 4 B and Supplementary figure 5A**)
- vii) **systematic quantification of transgene integration by on target ddPCR** in edited HSPCs (**new Supplementary figure 4C and Supplementary figure 5B**)

- viii) **systematic quantification of *HBA2* deletions** in edited HSPCs (Supplementary figure 1F, already present; **new Supplementary figure 4D and Supplementary figure 5C**)
- ix) **genotyping 69 BFU-E** for *HBA2* deletion and transgene integration (**new Supplementary figure 5E-F**)
- x) **analyzing 28 HUDEP-2 single cell clones to quantify *HBA1* vs *HBA2* targeted integration** (**new Supplementary figure 4A**)

We have added this information in the results and materials and methods sections of the revised manuscript.

We also tried to assess *HBA2* inversion in bulk population of HSPCs edited with *HBA15* RNP using 6 different PCR and we could not detect any specific amplification.

This is in line with a recent publication by Li et al. for the human γ -globin locus, which is similar to the α -globin one for the presence of evolutionary duplicated genes (*HGB1/HBG2* vs *HBA1/HBA2*). Specifically, the authors performed CRISPR/Cas9 editing of the γ -globin locus in HSPCs and they could detect genomic deletion but not genomic inversion. (PMID: 29789357; Fig 7).

Although we are persuaded that the lack of genomic inversion in our edited HSPCs is a true data, we decided not to insert this information in the revised version of the manuscript since we could not validate our PCR primers using an artificial template as positive control.

The reason is twofold:

- 3) Two manufacturers independently confirmed that the presence of repetitive sequences and the high GC content of the α -globin locus (PMID: 11157800) makes the DNA synthesis of such template not possible with current technology.
- 4) The forced lockdown of the lab for the current coronavirus pandemic does not allow us to perform any additional attempt for an unknown amount of time (probably 2 months).

If necessary, we can explain in the manuscript that we did not detect any *HBA2* inversion by PCR analysis, although we cannot formally exclude a false negative result.

“Along these lines, the ddPCR analysis of copy number (Fig. S1f) seems to show only a loss of one copy. Moreover, the data in Figure 3b is used to show correlation with GFP+ cells. However, HBA genes are both duplicated and autosomal (4 copies) what is the expected outcome of hetero and homozygous

targeting? The authors completely ignore this possibility. Please clarify, and provide supporting data.”

We thank the referee for the useful comment. We previously quantified the number of donor template targeted integration and **we observed mostly 1 integration per cell, less frequently 2 integration (Figure 3D and K, Figure 4H). We never observed 3 or 4 integrations**, which are either absent or too rare to be detected with our quantification methods.

In addition, to better characterize the outcome of the RNP-AAV treatment at single cell level we have:

- **Genotyped 69 single BFU-E** in term of HBA2 deletion and transgene (*F9* or *LAL*) integration (**new Supplementary Figure 5E-F**)
- Assessed that in **28 HUDEP-2 cell clones** transgene integration is occurring mostly in HBA1 gene (**new Supplementary Figure 4A**). Of 5 clones with double integration, 3 had transgene integration in the both HBA1 alleles and 2 in both HBA1 and HBA2 alleles.

“The authors use GFP to enrich cells with vector integrations. What is the overall frequency of integration (ie. %GFP+). It seems the data is only shown for Factor IX integration. (Fig 3b Pre Sort). Please include this data to demonstrate the overall targeting efficiency.

In the long run, is GFP a viable solution for therapeutic application? What is a reasonable alternative? Please discuss.”

We thank the referee for the useful comment. GFP sorting was performed only for HUDEP-2 cells (Figure 3B). In HSPCs experiments, transgene targeted integration was so efficient that no enrichment was required. **We have now specified this in the revised manuscript.**

“The authors screen multiple gRNAs targeting the 5’UTR or introns, and select two sites as leads.

However, the overall lack of detail on the homology arms, gRNAs and genotyping assays (apart from supplemental data) makes it difficult to understand the exact integration site chosen, or the rationale.”

We thank the referee for the useful comment. **We have now color-coded the gRNAs and we show their genomic location (Figure 1A-B and supplementary Figure 2 E-F). We have also added the genomic coordinates for the homology arms** used for HDR in the materials and methods section

“Homology arms for 5’UTR α -globin integration are: upstream, chr16:172,642-172,892; downstream, chr16:172,893-173,142 (hg38)

Homology arms for IVS2 α -globin integration are: upstream, chr16:173,135-173,385; downstream, chr16:173,386-173,636 (hg38)”

Moreover, the IVS2 integrations site and trap vector should produce a truncated α -globin protein, which is not screened for (eg. Western blot) or even discussed by the authors.”

We have now added this information in the result section of the revised manuscript:

“For this reason and considering that DNA targeted integration in IVS2 could result in the expression of a truncated α -globin chain, we selected the 5’UTR region for further investigation.”

“There are many instances of “data not shown”, against Nature policy. Also, colony screening data is not shown for Figure 3k, although HDR rates are reported (7/7).”

We have now added the missing data.

We now show the PCR for HDR for 19 BFU-E colonies (new Supplementary Figure 4L).

“The screening process for gRNAs is not clear. In Figure 1a – does each grey bar represent a different gRNA? A map of the locus (found in Fig. S1a) or appropriate labelling is required in the main figure.”

We added in the Figure 1 legend that: *“Each bar is a different gRNA, each dot a different experiment.”*

We have moved Supplemental Figure 1A to Figure 1A.

“Throughout the paper, the gRNA used for each experiment is not clear.

Which guides were chosen from Supp2e/f to produce the data in Panel g? (HBB targeting)”

We thank the referee for the useful comment. We have now color-coded the gRNA and we show their genomic location (Figure 1A-B and supplementary Figure 2E-F).

“Moreover, the details of the target site are not clear. While the 5’UTR gRNA cuts upstream of the ATG, does the targeting vector correct the position of the insertion to be in-frame with the endogenous ATG, or does it insert directly at the CRISPR/Cas9 cut site?”

We thank the referee for the useful comment. The targeting vector correct the position of the insertion to be in frame with the endogenous ATG.

In the materials and methods paragraph, we have now added: *“Upon successful HDR, transgene translation starts from the same ATG as the endogenous α -globin for 5’ UTR integration or after translation of a fragment of α -globin chain for IVS2 integration (α -globin-2A-GFP).”*

“Genotyping data for insertion at HBB is not shown.”

We have added the gel of the PCR analyses to confirm on-target integration in the HBB gene (Supplementary figure 2I-J)

“in line with TIDE-based indels analysis (8.5%±4.9, n=5)”

The TIDE data is not shown?”

We have added the data in Supplemental figure 3A.

“Remarkably, 5’UTR and IVS2 gRNA did not modify α -globin expression, measured as ratio between α - and β -like globin chains (Figure 1e and supplementary figure 1d).

Was modification expected? Did the authors make an effort to map or avoid regulatory elements? What was the nature of the indels formed? The sequence analysis of indels is not made available.”

gRNA targeting both α - and β -globin genes were designed to avoid known regulatory elements.

We have now specified this in the result paragraph: *“We designed 14 guide (g)RNA targeting the non-coding sequences of α -globin genes, in particular the 5’ untranslated region and introns (5’UTR, IVS1 and IVS2 respectively)¹⁸, avoiding known regulatory elements”.*

We have added the InDels pattern for both α -globin gRNA (5’UTR and IVS2, Supplementary Figure 1A) and β -globin gRNA (5’ UTR and IVS2, Supplementary Figure 2G)

“In the first set of results, the authors state “without affecting HSPC viability, differentiation potential and hemoglobin expression, thus representing ideal genomic sites for KI.”, However these data do not yet demonstrate transgene expression, and therefore do not support ideal sites for KI. Please re-write.”

We rewrote the sentence according to referee suggestion: *“...thus representing an interesting safe genomic locus for testing KI”*

“What are the predicted PCR fragment sizes in S2c/d? Presumably the PCR fragments differ in sizes amongst the clones because of NHEJ insertion of the IDLV. Please explain.”

We thank the referee for the useful comment. We specified in the result paragraph that: *“on-target integration by **non-homologous end joining** was confirmed in GFP positive clones by PCR...”*

“Do the two bars in Fig S1b represent two independent experiments, or two different gRNAs?”

We have now specified in the figure legend that:

“Each bar is a different gRNA plasmid transfection, each dot a different analysis”

“Lanes are not labelled in Fig S2i, and there is no schematic for the PCR assay.”

Each lane is a single colony; **we specify this in the figure legend (now Supplementary Figure S2L).**

We added the schematic for the PCR assay.

“KI occurred mostly through HDR (>90% of CFC, n=40; supplementary fig 2j).”

Fig S2j shows one sequencing result, not a summary of targeting efficiencies.”

The sequence represented in the figure (now Supplementary Figure 2m) is the sequence of 36 clones analyzed. Unfortunately, at the moment we cannot retrieve the other 4 sequences from the lab.

If necessary, we will remove this panel.

“The terms BFU-E, CFU-E, and CFU-GEMM are used in the text and need to be defined there. Is CFU-GM in Fig S2i the same as CFU-GEMM? The authors must be consistent.”

CFU-GM is not the same as CFU-GEMM. We now defined each abbreviations in the result paragraph and in the figure legends.

“PCR analysis of individual CFC showed integration in both red and white colonies (Supplementary figure 2j).” What is meant by red and white?”

We thank the referee for the useful comment. **We now wrote erythroid (red) and granulocyte-monocyte (white) colonies.**

“Figure 2h, please use scale bars, not magnification. Also, why are only positive cells shown? Including examples of GFP negative cells would be more convincing.”

We have replaced magnification with scale bars.

We have example of GFP negative clones (in the image below there is a GFP positive BFU-E next to a negative); however, since our targeting efficiency is not 100%, showing some GFP negative colonies will not provide any additional information.

“no indels were observed at any of the predicted off-target sites (Supplementary table 2).”

If that is the case, then what is the data in the table (column 4, last two columns)?”

We thank the referee for the useful comment. The last 2 columns of the Supplementary table 2 are the detected indel %.

We could not detect any difference in the levels of indels detected between HAB15 and control AAVS1 gRNAs; **we rewrote the result paragraph accordingly.**

The **TIDE software**, which was used to analyze the Sanger sequencing results of the PCR for the predicted off-targets, **is not reliable below ~2%** (PMID: 25300484)

Both information have been added in the revised version of the manuscript.

“Although on-target activity reached >85%, we could not detect any difference between HBA15 and control AAVS1 gRNA at any of the predicted HBA15 off-target sites (Supplementary table 2) (with a technical threshold of >2% of TIDE software).”

“Off-target candidates were predicted in silico using two different software”

Please state the software or provide a URL.

We have added the names of the two software in the materials and methods paragraph.

“BFU-E derived erythroblasts were capable of secreting FIX (supplementary figure 3c).”

Should the comment cite Fig. S4d?

We thank the referee for pointing out this error, which has been now corrected.

Reviewers' Comments:

Reviewer #1:

Remarks to the Author:

The Authors well addressed all my questions and the manuscript has been improved. Since inversions are a common outcome when cleaving two adjacent loci with nucleases and the Authors cannot rule out this possibility with a low false negative confidence, I would just recommend them to indicate in the manuscript that "the detection and quantification of HBA2 inversions was not possible due to technical issues".

Reviewer #2:

Remarks to the Author:

Considering circumstances, the authors have made a commendable effort to address the reviewer comments, overall improving the manuscript. The design and results are now much clearer. Still, concerns over potential genomic rearrangements - shared between both reviewers - remain unresolved. Instead, the authors provide published examples of comparable gene editing scenarios. While this is not direct evidence, it is understood that analysis of rearrangements in duplicated genomic regions can be complex. Therefore, the authors must make it clear in the discussion that their data cannot exclude this potential outcome of their editing strategy. Regarding the sequence data in Supplementary Figure 2m, the authors may have misunderstood my concern. The results state a percentage editing, while the data shown is not the HDR results of 40 clones, just PCR products for 14 samples and one sequence read. I understand that they may have difficulties accessing their lab, however the conclusion must be supported by appropriate data (for example, a table of clonal sequencing results).

Reviewer #1 (Remarks to the Author):

The Authors well addressed all my questions and the manuscript has been improved. Since inversions are a common outcome when cleaving two adjacent loci with nucleases and the Authors cannot rule out this possibility with a low false negative confidence, I would just recommend them to indicate in the manuscript that “the detection and quantification of HBA2 inversions was not possible due to technical issues”.

We thank the referee for the useful comments that helped improving the manuscript.

We have now added this point in the results paragraph of the revised version of the manuscript:

“Detection and quantification of HBA2 inversions was not possible due to technical issues associated with the presence of repetitive sequences and the high GC content of the α -globin locus.”

Reviewer #2 (Remarks to the Author):

Considering circumstances, the authors have made a commendable effort to address the reviewer comments, overall improving the manuscript. The design and results are now much clearer.

We thank the referee for the useful comments that helped improving the manuscript.

Still, concerns over potential genomic rearrangements - shared between both reviewers - remain unresolved. Instead, the authors provide published examples of comparable gene editing scenarios. While this is not direct evidence, it is understood that analysis of rearrangements in duplicated genomic regions can be complex. Therefore, the authors must make it clear in the discussion that their data cannot exclude this potential outcome of their editing strategy.

We have now added this point in the results paragraph of the revised version of the manuscript:

“Detection and quantification of HBA2 inversions was not possible due to technical issues associated with the presence of repetitive sequences and the high GC content of the α -globin locus.”

Regarding the sequence data in Supplementary Figure 2m, the authors may have misunderstood my concern. The results state a percentage editing, while the data shown is not the HDR results of 40 clones, just PCR products for 14 samples and one sequence read. I understand that they may have difficulties accessing their lab, however the conclusion must be supported by appropriate data (for example, a table of clonal sequencing results).

In order to address reviewers concern, we now only focus on the 10 KI clones for which we can provide the PCR image. For all of them we got seamless HDR integration, resulting in the same Sanger sequencing profile (shown in Sup Fig 2m).

We now edited the text as follow:

“Sanger sequencing of PCR products spanning the AAV-genome junction of colonies showed that KI occurred through HDR (n=10; Supplementary Figure 2l, m)”